# *Drosophila* nicotinic acetylcholine receptor subunits and their native interactions with insecticidal peptide toxins

Dagmara Korona[1†], Benedict Dirnberger[1,2,3*†], Carlo NG Giachello[3†],
Rayner ML Queiroz[2], Rebeka Popovic[4], Karin H Müller[5], David-Paul Minde[2],
Michael J Deery[2], Glynnis Johnson[1], Lucy C Firth[3], Fergus G Earley[3],
Steven Russell[1], Kathryn S Lilley[6]

[1]Department of Genetics, University of Cambridge, Downing Street, Cambridge, United Kingdom; [2]Cambridge Centre for Proteomics, Department of Biochemistry, University of Cambridge, Cambridge, United Kingdom; [3]Syngenta, Jealott's Hill International Research Centre, Bracknell, United Kingdom; [4]MRC Toxicology Unit, Gleeson Building, University of Cambridge, Tennis Court Road, Cambridge, United Kingdom; [5]Cambridge Advanced Imaging Centre, Department of Physiology, Development and Neuroscience/Anatomy Building, University of Cambridge, Cambridge, United Kingdom; [6]Cambridge Centre for Proteomics, Department of Biochemistry, University of Cambridge, Tennis Court Road, Cambridge, United Kingdom

*For correspondence:
bd415@cam.ac.uk

†These authors contributed equally to this work

**Abstract** *Drosophila* nicotinic acetylcholine receptors (nAChRs) are ligand-gated ion channels that represent a target for insecticides. Peptide neurotoxins are known to block nAChRs by binding to their target subunits, however, a better understanding of this mechanism is needed for effective insecticide design. To facilitate the analysis of nAChRs we used a CRISPR/Cas9 strategy to generate null alleles for all ten *nAChR* subunit genes in a common genetic background. We studied interactions of nAChR subunits with peptide neurotoxins by larval injections and styrene maleic acid lipid particles (SMALPs) pull-down assays. For the null alleles, we determined the effects of α-Bungarotoxin (α-Btx) and $\omega$-Hexatoxin-Hv1a (Hv1a) administration, identifying potential receptor subunits implicated in the binding of these toxins. We employed pull-down assays to confirm α-Btx interactions with the *Drosophila* α5 (Dα5), Dα6, Dα7 subunits. Finally, we report the localisation of fluorescent tagged endogenous Dα6 during *Drosophila* CNS development. Taken together, this study elucidates native *Drosophila* nAChR subunit interactions with insecticidal peptide toxins and provides a resource for the in vivo analysis of insect nAChRs.

## Editor's evaluation

The authors employ genetic and biochemical approaches to demonstrate the insecticidal effects of a snake peptide toxins. Intriguingly, they show that it targets different nicotinic acetylcholine receptor subunits than a previously identified insecticidal spider toxin. Especially their clever combination of detergent-free membrane protein extraction and mass spectrometry will no doubt prove useful to study native receptor-ligand interactions in the future.

## Introduction

Global climate change and other factors are placing increasing demands on available agricultural land to deliver efficient, reliable and sustainable food production. Insecticides are important tools in securing yields of all major crops but need to be continually replaced to overcome resistance in target species and reduce environmental impacts. In addition, new insecticides must have low toxicity to non-target species, particularly the major pollinators essential for agriculture. A large class of insecticide targets are neurotransmitter receptors such as the nicotinic acetylcholine receptors (nAChRs) located in synaptic plasma membranes (*Ihara et al., 2020*). These pentameric cys-loop ligand-gated ion channels consist of either only α-subunits or α- and β-subunits, with ligand-binding sites located between two α-subunits or between α- and β-subunits. Most insect genomes, including that of the highly tractable *Drosophila melanogaster* model, harbour 10 highly conserved subunit genes that assemble in various combinations to form the active receptors.

An essential pre-requisite for effective design of new insecticides targeting these receptors is an understanding of their distinctive binding properties. For many reasons, including low expression in endogenous tissues or difficulties in expressing insect receptors in heterologous systems, the characterisation of functional insect receptors has been challenging (*Perry et al., 2021*; *Zuo et al., 2022*; *Salgado, 2021*). Even in the tractable *D. melanogaster* insect model, there has been no systematic isolation of mutations in *nAChR* subunit genes, until recently, when Perry and colleagues described the generation of a new set of null mutations in nine out of the ten *D. melanogaster* subunit genes (*Perry et al., 2021*). These mutations, however, were generated in different genetic backgrounds necessitating additional work to assay background sensitive phenotypes such as neural or behavioural defects.

Several classes of insecticide, the most effective being those in the neonicotinoid and spinosad class, have been shown to bind insect nAChRs highly selectively to block their functions (*Chambers et al., 2019*; *Houchat et al., 2019*; *Martelli et al., 2022*; *Lu et al., 2022*). Recently, the binding affinity and the positive allosteric effects of ω-Hexatoxin-Hv1a (Hv1a) peptide on nAChRs has been demonstrated (*Chambers et al., 2019*) and this spider venom peptide is well known for its insecticidal effects. In addition, other peptide toxins, such as the snake venom constituent, α-Bungarotoxin (α-Btx), have been widely used to probe nAChR functions, however whether α-Btx harbours a selective insecticidal property is currently unknown. Alpha-Btx is a 74 amino acid peptide that binds irreversibly to nAChR α-subunits in different species, including *D. melanogaster,* although the exact binding targets of these receptors are not fully understood (*Schmidt-Nielsen et al., 1977*; *Dellisanti et al., 2007*; *daCosta et al., 2015*). Landsdell and co-workers have shown binding of α-Btx to *D. melanogaster* α5 (Dα5), Dα6, and Dα7 subunits in a heterologous S2 cell expression system (*Lansdell and Millar, 2004*; *Lansdell et al., 2012*). The amino acid sequences of these subunits show strong similarity across their ligand-binding domains (LBD), suggesting that at least three subunits with sequence similarity are implicated in $\alpha$-Btx binding in vivo.

The lipid bilayer surrounding nAChRs is known to be essential for structural integrity, stability, and ligand-binding (*daCosta et al., 2013*). However, this lipid requirement can make analysis of membrane protein complexes challenging. The development of methods for extracting membrane proteins from lipid bilayers using detergents and introducing them into artificial lipid nanodiscs has facilitated a much better characterisation of receptor-ligand interactions (*Denisov and Sligar, 2016*). The use of detergents generally employed to solubilise membrane proteins, however, leads to destabilisation, aggregation and misfolding and their use is therefore not compatible with this type of analysis (*Loo et al., 1996*). Styrene maleic acid lipid particles (SMALPs) allow detergent-free extraction of membrane proteins in their local lipid environment and provide a promising technique for investigating receptor-ligand interactions under native conditions (*Lee et al., 2016*). This is particularly important since loss of lipids surrounding membrane proteins can lead to changes in measured binding affinities (*Martens et al., 2018*; *Gault et al., 2020*). The combination of detergent free SMALPs extraction coupled with mass spectrometry analysis provides a potential route for characterising native membrane receptor complexes (*Sobotzki et al., 2018*; *Kalxdorf et al., 2021*).

Here, we report the results from a combined genetic and biochemical analysis of *D. melanogaster* nAChRs in vivo. Using CRISPR/Cas9 genome engineering, we generated new null mutations for all ten receptor subunit genes in a uniform genetic background as well as introducing a fluorescent protein tag into the *nAChRα6* locus. We show that the null mutants in all seven α-subunit genes and two of the

three β-subunit genes are viable and fertile, although we find mild morphological defects and some neurological impairment. Mutation of the remaining subunit gene, *nAChRβ1*, is recessive lethal. All nine of the viable null mutants were used to demonstrate a novel selective insecticidal effect of α-Btx on the *nAChRα5, nAChRα6 and nAChRα7* subunits. We also applied the insecticidal Hv1a peptide to the viable null mutants, showing resistance with two subunit gene mutants: *nAChRα4 and nAChRβ2*. In our biochemical studies, we analysed receptor-ligand interactions in native conditions using SMALPs to verify the in vivo receptor subunit α-Btx binding targets in adult neural tissue from wild-type and receptor subunit mutants. Our analysis revealed binding of α-Btx to receptors containing Dα5, Dα6 and Dα7 subunits with the analysis of mutants in these subunit genes indicating heterogeneity in α-Btx binding nAChRs. Furthermore, we have identified specific glycosylation sites in Dα5 and Dα7 subunits which are known from other studies to play a critical role in α-Btx-binding affinity (*Dellisanti et al., 2007*; *Rahman et al., 2020*). Localisation studies with the Dα6 subunit tagged at the endogenous locus with a fluorescent reporter shows expression at different developmental stages in specific neuronal cells, including the Kenyon cells of the mushroom bodies, a known site of α-Btx-binding (*Su and O'Dowd, 2003*).

## Results

### Construction of new *D. melanogaster* nicotinic acetylcholine receptor subunit gene mutations

To investigate the role of individual nAChR subunits we used CRISPR/Cas9 genome engineering to generate null mutations in each of the seven α-subunit and three *β*-subunit genes. All mutations were generated in virtually identical genetic backgrounds using *nos*-Cas9 sources on the second or third chromosome of otherwise genetically homogeneous fly lines. Two of the subunit genes, *nAChRα3* and *nAChRα7*, are X-linked and consequently these lines retain the X from the nos-Cas9 stocks used to generate the mutations whereas the other eight null mutations have a *w^1118*^ chromosome. In brief, for each gene we targeted exons shared between all predicted isoforms, in most cases close to the N-terminus of the protein. To disrupt each coding sequence and facilitate screening we introduced a visible fluorescent marker, DsRED under control of the eye-specific 3xP3 promoter at the targeted locus (*Figure 1—figure supplement 1A-C*). Positive lines were confirmed by PCR and sequencing. The system was designed in such a way that the DsRED marker may be excised from the genome by Cre-LoxP recombination, leaving a 3 X FLAG tag and a single LoxP site with the reading frame predicted to be restored. We have not yet tested this facet of the design.

For nine out of 10 subunit genes, we established homozygous viable and fertile stocks, the exception was the *nAChRβ1* gene which proved to be recessive lethal. We crossed the *nAChRβ1* null strain with a *Tubby^1^*-marked balancer but obtained no homozygous 3rd instar larvae, indicating lethality prior to this stage, which precluded subsequent analysis of homozygotes. Although all the other lines are viable, we noticed that all of the lines, but particularly *nAChRα1, nAChRα2, nAChRα5* and *nAChRβ3*, exhibited a curled abdomen phenotype that is most prominent in males (approximately 25, 20, 15% and 15% respectively, *Figure 1A and B*, *Supplementary file 1*).

It is possible that this phenotype is a result of defects in neural control of abdominal muscles and it is interesting to note that a previous analysis of an *nAChRα1* allele reports reduced male courtship and mating (*Somers et al., 2017*). Since nAChRs are mostly found in the nervous system, we carried out basic climbing assays on the null alleles to assess potential locomotor defects (*Figure 1C*, *Supplementary file 2*). We saw little or no impact on the locomotor activity of 10-day-old flies with *nAChRα4, nAChRα5, nAChRα7, nAChRβ2*, or *nAChRβ3* homozygous mutants.

In contrast, the *nAChRα1, nAChRα2,* and *nAChRα6* mutants showed 50–60% reductions in climbing ability compared to wild-type, and the *nAChRα3* null mutant exhibited a severe reduction in locomotor activity with less than 25% of wild-type activity. Taken together, we report the generation and validation predicted of null mutations in all 10 *D. melanogaster* nAChR subunit genes, with mild morphological defects associated with most of the new alleles and impaired locomotion observed with some of the mutants.

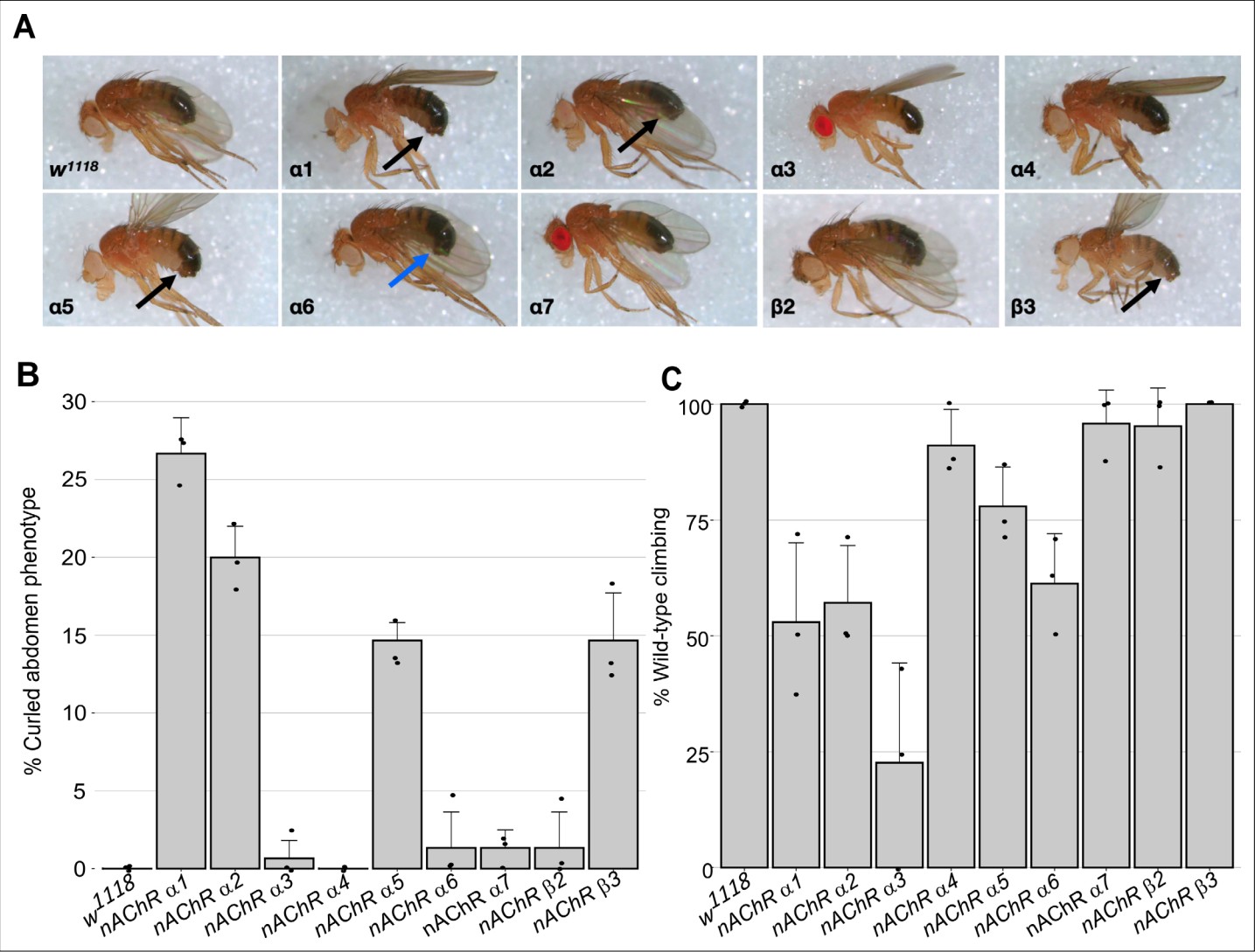

**Figure 1.** Morphological and locomotor phenotypes in *nAChR* subunit mutants. (**A**) Adult males from indicated *nAChR* subunit null mutants, black arrows indicate curled abdomens but even in lines with a low frequency the phenotype is prominent (blue arrow). (**B**) Frequency of curled abdomen phenotype (%), n=3. (**C**) Graph of locomotor activity determined in climbing assays as a percentage of wild-type, n=3.

The online version of this article includes the following figure supplement(s) for figure 1:

**Figure supplement 1.** Construction of new *D. melanogaster* nicotinic acetylcholine receptor subunit gene mutations.

## Distinct nAChR subunits mediate interactions with Ω-Hexatoxin-Hv1a and α-Bungarotoxin

In order to investigate the selective contribution of each *nAChR* subunit to toxin binding in vivo, we injected 3rd instar larvae from the homozygous *nAChR* mutants with either ω-Hexatoxin-Hv1a (Hv1a) or α-Bungarotoxin (α-Btx) dissolved in PBS. As a control, injections of PBS alone (vehicle) were performed in parallel, with all larvae surviving the injection procedure and showing no detectable defects. Larval injection of 2.5 nmol/g Hv1a induced locomotor paralysis and full lethality in the control groups ($w^{1118}$, *THattP40* and *THattP2*, *Figure 2A*, *Supplementary file 3*). Survival was quantified as the percentage of pupae formed after injection. Injection of Hv1a into each of the receptor mutant lines showed varied effects. Where *nAChRα3*, *nAChRα5*, *nAChRα6*, and *nAChRα7* showed lethality comparable to the wild-type controls ($p > 0.9$), the *nAChRα4* and *nAChRβ2* mutants showed a significant increase in survival to 42% ± 22% (one-way ANOVA followed by Bonferroni's test, $p = 0.0035$, *Figure 2A*). *nAChRα1*, *nAChRα2* and *nAChRβ3* showed slight increases in survival but these were not found to be significant.

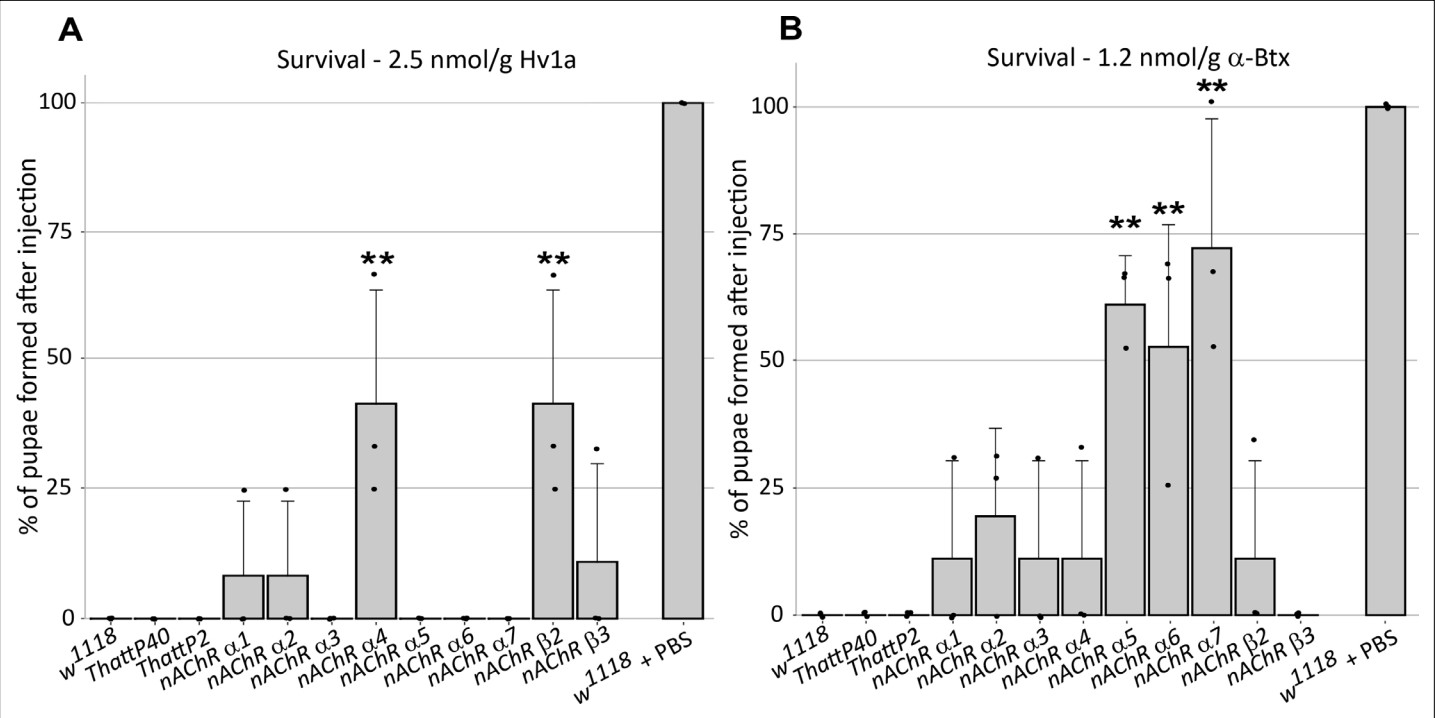

**Figure 2.** ω-Hexatoxin-Hv1a and α-Bungarotoxin target different nAChR subunits. (**A**) Bar graph of the survival rate, measured as the percentage of pupae formed, following larval injection of 2.5 nmol/g Hv1a in the indicated homozygous lines. **p=0.0035 (one-way ANOVA F(11,24) = 4.99, p=0.0005 with Bonferroni's multiple comparisons test). Mean ± SD of three independent replicates in each group (10 injected larvae in total). Individual replicate data points are shown (**B**) Survival rate following larval injection of 1.25 nmol/g α-Btx. **p<0.001, ***p=0.0001, one-way ANOVA (F(11,24) = 7.921, p<0.0001, followed by Bonferroni's multiple comparisons test). Mean ± SD of three independent replicates in each group (10 injected larvae in total). *w1118* is the wild-type base stock, *THattp40* and *THattP2* are the *Cas9* lines used to establish the mutants, *w1118* + PBS represents the injection control.

We also observed significant toxicity following injection of 1.25 nmol/g α-Btx, with larvae exhibiting a progressive reduction in locomotion until stationary, resulting in developmental arrest and death. We found that α-Btx induced lethality is drastically reduced in the *nAChRα5*, *nAChRα6* and *nAChRα7* subunit mutants, with the survival rate significantly increased from 0% (controls) to 61% ± 10% (p = 0.001), 53% ± 24% (p = 0.0051) and 72% ± 25% (p = 0.0001), respectively (one-way ANOVA followed by Bonferroni's test, *Figure 2B*). Together, these results indicate that Hv1a and α-Btx do not share the same binding target and differentially interact with the nAChR subunits in vivo. Since α-Btx showed a novel insecticidal effect on nAChRs we further examined its interactions biochemically.

## Forming SMA-lipid particles (SMALPs) of ring-like nAChR complex structures

To take advantage of our novel receptor mutants for the characterisation of native nAChR functions, we undertook a biochemical analysis to determine the subunits responsible for binding α-Btx. The analysis of membrane proteins such as nAChRs and their native in vivo interaction with toxins by mass spectrometry has proved extremely challenging (*Mulcahy et al., 2018*) since effective solubilisation of native receptors in an environment that maintains subunit interactions and receptor integrity is important for characterising toxin-receptor binding.

We reasoned that solubilisation of low abundance nAChR complexes into styrene maleic acid lipid particles should facilitate characterisation of native interactions with α-Btx. We therefore utilised detergent-free SMALPs extraction to characterise the interaction between receptor native lipid discs and the α-Btx toxin (*Figure 3A*).

In brief, we prepared membrane extracts from adult *D. melanogaster* heads (*Depner et al., 2014*) and generated lipid particle discs by solubilising the membrane extracts with the SMA copolymer. We used affinity beads coupled to α-Btx (*Wang et al., 2003*; *Mulcahy et al., 2018*) to enrich for nAChRs in the SMALP preparations that bound to the toxin, and performed mass spectrometric analysis of

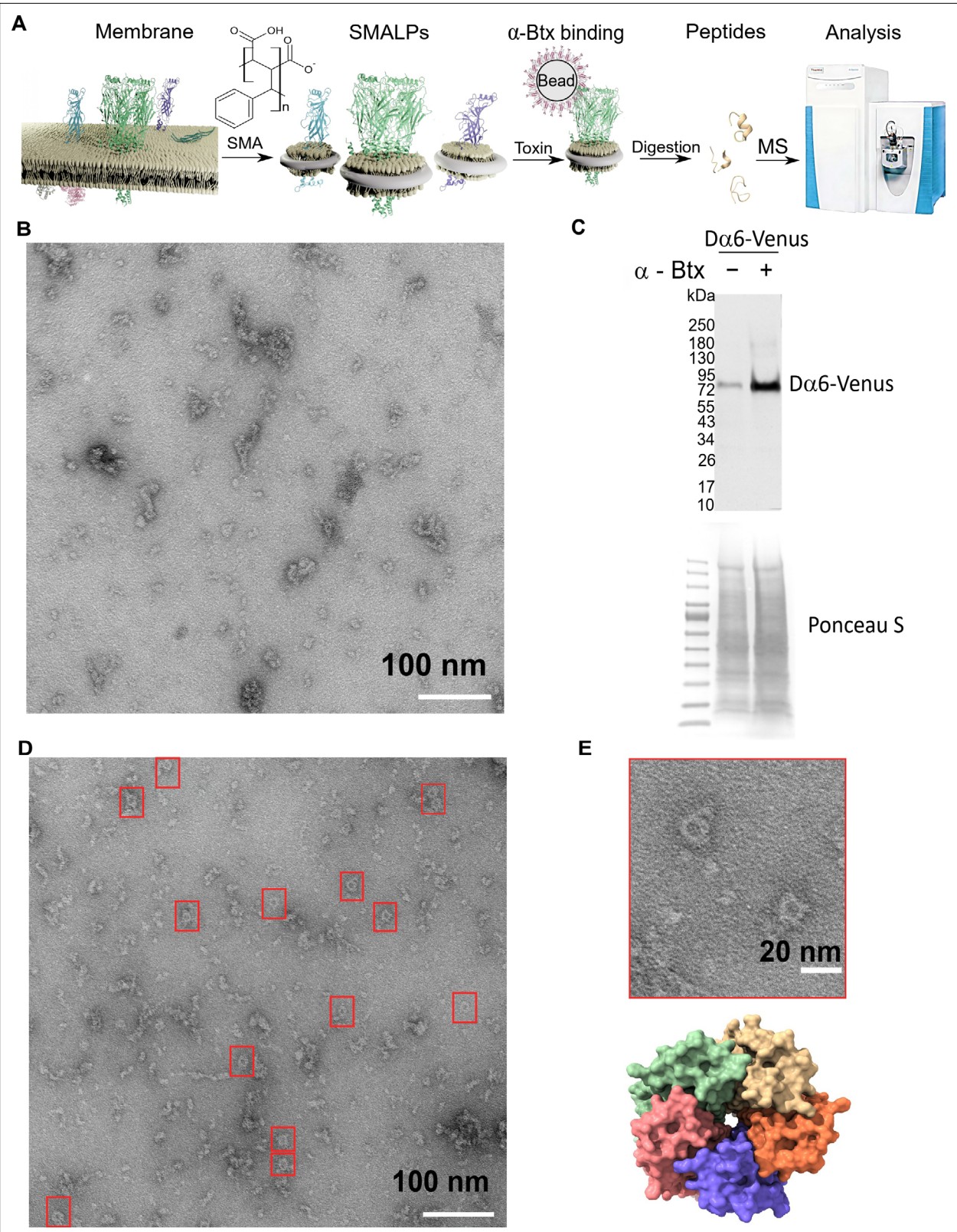

**Figure 3.** Forming styrene maleic acid lipid particles (SMALPs). (**A**) Schematic representation of the SMALPs extraction and nAChRs pull-down for mass spectrometric analysis. (**B**) Negative staining of extracted SMALPs by transmission electron microscopy, n=3. Scale bar 100 nm. (**C**) Western blot for Dα6-mVenus nAChR with and without enrichment using α-Btx, n=2. Detected with anti-GFP antibody. The fusion protein was detected at approximately 83 kDa. Ponceau S staining was used as sample equal loading control. (**D, E**) Negative staining of extracted SMALPs after α-Btx pull-downs, n=3, ring-like

*Figure 3 continued on next page*

*Figure 3 continued*

protein structures are boxed (scale bar = 100 nm) with an example in the magnified image (scale bar = 20 nm). A top view of the nAChR structure from PDB entry 4HQP is shown for reference.

The online version of this article includes the following source data and figure supplement(s) for figure 3:

**Source data 1.** Transmission electron microscopy micrographs.

**Figure supplement 1.** Coupling α-Btx and testing pull-down efficiency with affinity beads.

**Figure supplement 1—source data 1.** Coupling efficiency of α-Btx to affinity beads.

tryptic peptides generated from the enriched preparations. In parallel, we processed membrane extracts without SMALP and with SMALP extracts enriched with beads alone as controls.

We first determined whether membrane protein discs are formed from enriched membranes using the SMA copolymer. We prepared membrane enriched fractions from adult heads, solubilised these with SMA and separated the insoluble particles from the lipid discs by ultracentrifugation. We negatively stained the SMALP preparations and imaged them with transmission electron microscopy (TEM), observing irregular discs of varying shapes and sizes, with clusters containing different numbers of discs (n = 3, 13 micrographs are provided, *Figure 3B*). Membrane receptors often have a unique shape in TEM images and the pentameric nAChR is expected to form a ring-like structure, suggesting that the receptors are extracted as a complex. However, we did not observe ring-like structures, suggesting that nAChRs are of low abundance and that analysis may benefit from enrichment. We thus coupled α-Btx to affinity beads to enrich nAChR complexes that bind the toxin in SMALP preparations. We applied a C-terminal mVenus tagged Dα6 nAChR subunit strain to verify the presence and the enrichment of this fusion protein via Western blotting using an anti-GFP antibody (*Figure 3C*, *Figure 3—figure supplement 1A*). Pull-downs using α-Btx affinity resin showed strong enrichment of the Dα6-mVenus fusion protein at approximately 83 kDa compared to the unenriched control samples (*Figure 3C*, *Figure 3—figure supplement 1B*). A second band was detectable at approximately 180 kDa in pull-down samples without treatment with the reducing agent DTT, suggesting that in the pull-downs Dα6 is present as a dimer (*Figure 3—figure supplement 1C, D*). In contrast to the unenriched samples, TEM micrographs of the enriched preparations showed increased numbers of ring-like structures of 15 nm in diameter (n = 3, 15 micrographs are provided, *Figure 3D, E*).

Thus our TEM analysis shows an enrichment in ring-like membrane complexes in the SMALP preparations which are likely to be nAChRs.

## Efficient SMALPs extraction allow to study nAChR subunits solubility

To assess to what extent the SMA copolymer solubilised nAChRs, we performed a shotgun proteomics analysis to identify receptor subunits. Membrane preparations were solubilised in buffer with or without SMA, and affinity beads with or without α-Btx were used to assess binding to nAChR subunits. Comparing the number of proteins identified in samples solubilised either with or without 5% SMA, we observed a significantly increased identification of proteins dissolved in SMA by equal numbers of MS/MS spectral counts (two-tailed t-test, p < 0.01, *Figure 4A* and non-significant, *Figure 4B*). This indicates that mass spectrometer performance was comparable during the analysis.

Sequences of membrane spanning segments of nAChR subunits, which are embedded in a hydrophobic lipid environment, are largely composed of nonpolar side chains. Determining the average hydrophobicity of identified protein sequences revealed significantly increased numbers of proteins with a positive hydrophobicity score in samples solubilised in SMA (two-tailed t-test, p < 0.0001, *Figure 4C*), indicative of enrichment of membrane proteins. The hydrophobicity score was calculated as the sum of the hydrophobic or hydrophilic properties of amino acids divided by the length of identified proteins. An analysis of Gene Ontology (GO) slim terms supports the conclusion that the SMALP preparations are enriched for membrane embedded and/or associated proteins (*Figure 4D*), and that these are not limited to plasma membrane proteins. In the SMA-enriched samples, we found enrichment for proteins annotated with metabolic and catalytic activity terms and also enhanced response to biological stimuli (*Figure 4—figure supplement 1A, B*), highlighting the recovery of membrane-associated proteins. Next, we focused on identified membrane proteins predicted to contain transmembrane helical (TMH) domains and found an increased number of proteins containing TMHs in SMA solubilised samples

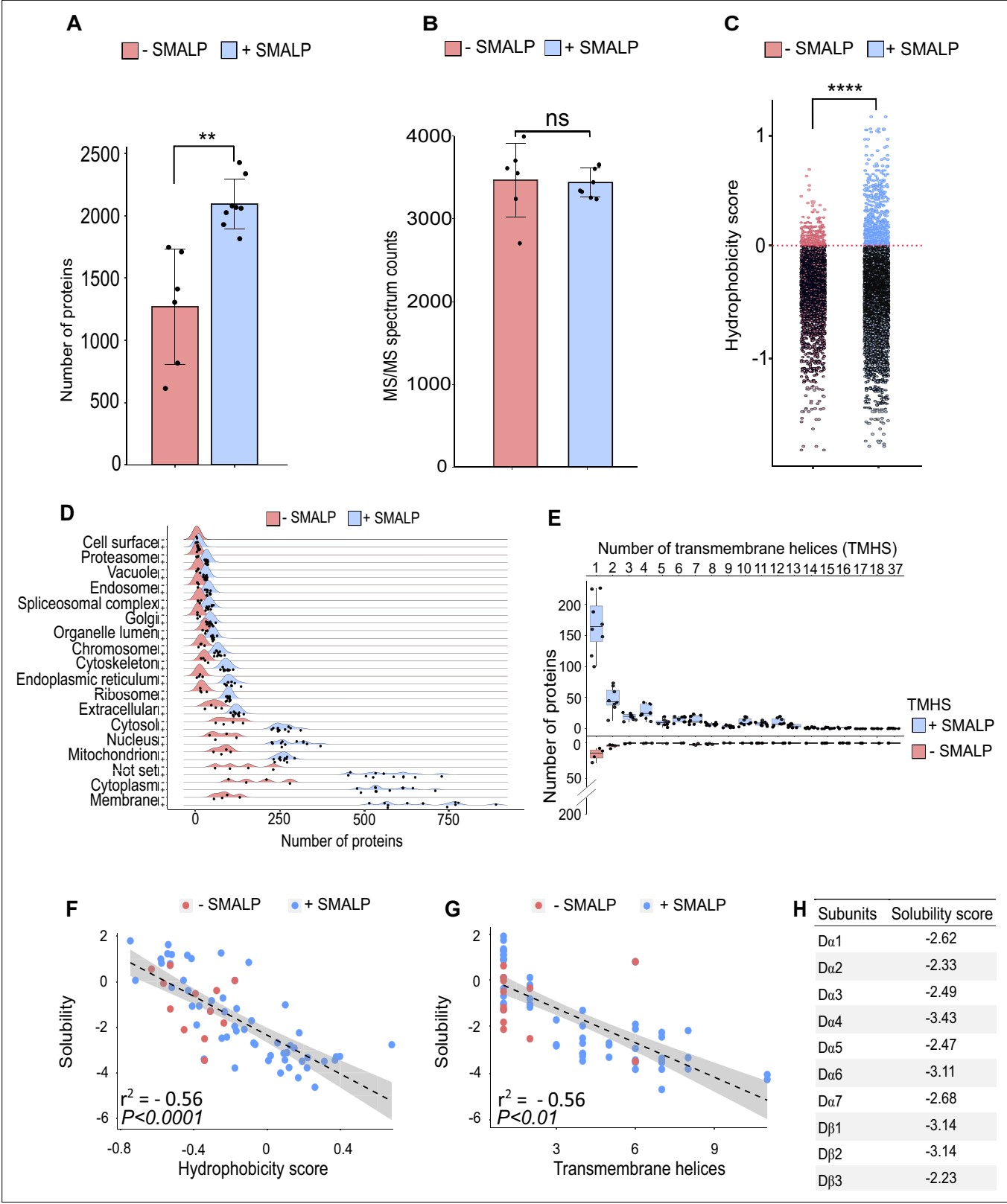

**Figure 4.** Identification of proteins enriched by SMALP extraction. (**A**) Number of identified proteins in affinity pull-down samples solubilised with or without SMA, two-tailed t-test, **p<0.01, n=6 or 8 replicates per condition. (**B**) MS/MS spectrum counts from samples solubilised with or without SMA, ns = not significant after two-tailed t-test with n=6 or 8. (**C**) Calculated hydrophobicity score of amino acid residues found in protein sequences obtained with and without SMA solubilisation, ****p<0.0001, two-tailed t-test, n=3 per condition. (**D**) GO term (cellular compartment) enrichment of proteins

*Figure 4 continued*

identified with and without SMA solubilisation, n=4 or 11. (**E**) Predicted numbers of proteins containing transmembrane helices obtained with or without SMA solubilisation, n=4 or 8. (**F, G**) Analysis of solubility and hydrophobicity of receptors identified with and without SMA solubilisation ($r^2$ = –0.56, p<0.0001, n=4) and of transmembrane receptor helices ($r^2$=0.56, p < 0.01, n = 4). (**H**) Solubility score of individual nAChR subunits.

The online version of this article includes the following source data and figure supplement(s) for figure 4:

**Source data 1.** SMALP extraction raw data.

**Figure supplement 1.** GO terms and predicted membrane proteins.

**Figure supplement 1—source data 1.** Further SMALP extraction raw data.

(*Figure 4E*). While the majority of these proteins contained a single TMH domain, we identified Piezo, a mechanosensory ion channel protein containing 37 predicted transmembrane helices. Both α- and β-nAChR subunits contain four TMH domains and could be solubilised in SMA. The number of β-barrel membrane spanning proteins identified was also significantly increased by SMA extraction (two-tailed t-test, p < 0.0001, *Figure 4—figure supplement 1C*). In addition, palmitoylated lipid anchor modifications to nAChR subunits has been shown to be important for receptor assembly into membranes and the formation of functional complexes (*Alexander et al., 2010*).

We found a significantly increased identification of proteins which are predicted to be palmitoylated and myristoylated (two-tailed t-test, p < 0.0001, *Figure 4—figure supplement 1D, E*). In contrast, membrane proteins that are predicted to contain a glycosylphosphatidylinositol (GPI)-anchor were equally solubilised in both conditions (two-tailed t-test, non-significant, *Figure 4—figure supplement 1F*). Focusing on the membrane receptors solubilised by SMA, we analysed the amino acid sequence properties of identified proteins and calculated an overall solubility score (*Sormanni et al., 2015*; *Sormanni et al., 2017*). Comparing the solubility to the hydrophobicity showed a calculated $R^2$ of 0.56 (*Figure 4F*). Sequences with a score greater than 1 are highly soluble and those less than –1 are difficult to solubilise. We therefore concluded that samples solubilised in SMA contain more receptors, which are difficult to solubilise. These receptors are more hydrophobic and contain larger numbers of TMH domains (*Figure 4G*). Calculating an average solubility score of –2.76 for nAChR sequences indicates that difficult to solubilise subunits are successfully recovered with SMA (*Figure 4H*).

Taken together, these data confirm that SMA solubilises nAChR complexes in a state suitable for subunit identification by mass spectrometry and suggests that α-Btx interactions can be studied with SMALP preparations.

## Three nAChR α-subunits are targets of α-Btx

Using nAChRs solubilised in SMA allowed us to search for native interactions between α-Btx and the receptor. To do so, we identified peptides from subunit ligand-binding and cytoplasmic domains, identifying the Dα5, Dα6, and Dα7 subunits in the α-Btx affinity bead pull-downs (*Figure 5A*, *Figure 5—figure supplement 1A*, *Supplementary file 4*). We also performed similar pull-downs with our newly constructed knockout mutants (*Figure 5B*). Several other nAChR subunit peptides could be identified in the negative controls performed without coupling α-Btx to affinity beads (*Supplementary file 5*). The sequences of the ligand-binding domains of the Dα5, Dα6, and Dα7 subunits are very similar (avg. 95.49 %) and we identified peptides common to all three subunits (*Figure 5—figure supplement 2A*) as well as unique peptides within each of cytoplasmic domains (*Figure 5—figure supplement 2B*). However, we found no evidence of peptides mapping to the TMH domains, which was not unexpected since peptides from these regions contain stretches of hydrophobic amino acids that are known to be difficult to detect by shotgun proteomics approaches (*Carroll et al., 2007*).

The ligand-binding domain of α-subunits show structural similarity across different species (*Figure 5C*) and by mapping the identified peptides to known subunit structures we concluded they were most likely outside of the α-Btx binding sites (*Figure 5D*). This may indicate that α-Btx binding sites are not structurally conserved or that binding of α-Btx alters local proteolytic susceptibility and as a consequence peptides identifiable by mass spectrometry are not generated.

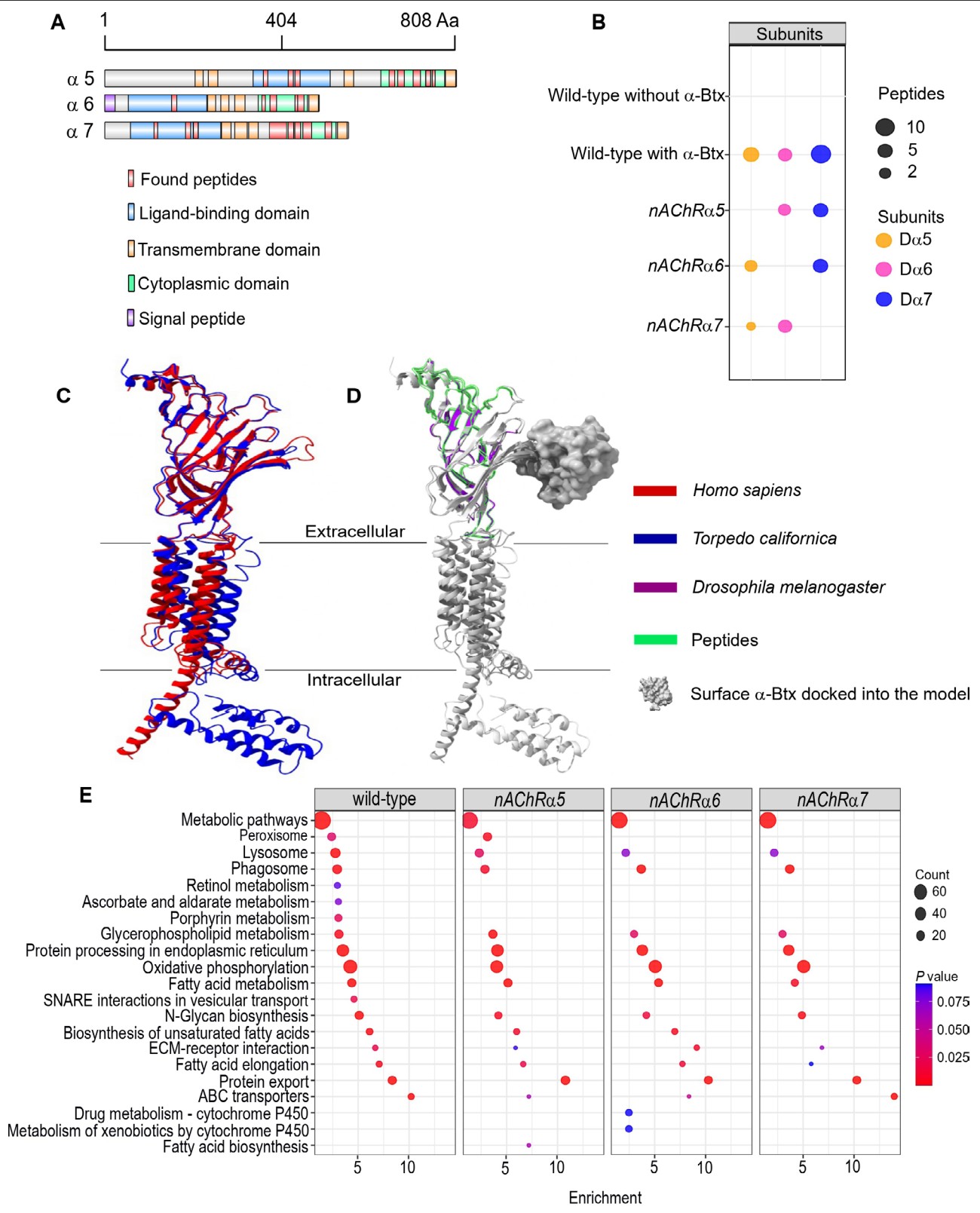

**Figure 5.** Three nAChR α-subunits bind to α-Bungarotoxin (α-Btx). (**A**) Identified Dα5, Dα6, and Dα7 subunit peptides in pull-downs using α-Btx affinity beads. Peptides from the ligand-binding and cytoplasmic domain are highlighted in red. (**B**) Numbers of identified unique peptides in wild-type pull-downs using affinity beads in absence and presence of α-Btx, n=3. Deleting *nAChRα5, nAChRα6, nAChRα7* and performing pull-downs identified unique peptides for nAChR subunits suggesting that functional complexes can be formed in each of the mutants, n=3. (**C**) Superimposed nAChR

*Figure 5 continued on next page*

Figure 5 continued

α-subunit structures from *Homo sapiens* (blue, PDB 6USF) and *Torpedo californica* (red, 6UWZ). The extracellular ligand-binding domains (LBD) exhibit a structure similarity. (**D**) Same superimposed structures docked to α-bungarotoxin (α-Btx, surface structure). Peptides found in LBD are highlighted in green. The homology regions of the Dα6 LBD are shown in violet.(**E**) KEGG pathway enrichment analysis of pull-downs in wild-type and *nAChRα5, nAChRα6, nAChRα7* mutants, Fisher's exact test, n=3. Protein counts with p values of enriched pathways are shown. p values of ≤ 0.05 are to be considered as strongly enriched with default threshold of 0.1.

The online version of this article includes the following source data and figure supplement(s) for figure 5:

**Source data 1.** Null pull-down raw data.

**Figure supplement 1.** Illustration of nAChR subunits.

**Figure supplement 2.** Identified peptides in ligand-binding and cytoplasmic domain.

To further characterise the role of the three α-subunits identified in as contributing to α-Btx binding we generated SMALP preparations and performed α-Btx affinity bead enrichments with adult head preparations from our homozygous mutants in the *nAChRα5*, *nAChRα6*, and *nAChRα7* subunit genes. With all three mutants we observed, as expected, no detectable peptides from the missing subunit but we still identified peptides from the other two subunits (*Figure 5—source data 1*).

We compared the repertoire of proteins identified with α-Btx enrichment in wild-type with those found in each of the three mutant lines to identify any changes in the representation of biological pathways annotated in KEGG (*Kanehisa et al., 2020*, *Figure 5E*). While the enrichments in wild-type and the mutants were broadly similar, we noticed a loss of proteins associated with cofactor/vitamin metabolism, particularly retinol and ascorbate, in all three of the mutants as well as proteins associated with vesicular transport. It is possible that these pathway changes represent alterations in neurotransmitter production or trafficking. Interestingly, we also noticed specific enrichment of cytochrome P450 related pathways in the *nAChRα6* mutants, suggesting perturbation of neurotransmitter pathways.

In summary, our analysis indicates that a functional α-Btx binding nAChR involves the Dα5, Dα6, and Dα7 subunits. This is entirely in line with our genetic findings described above, where loss of each of these subunit genes conferred substantial resistance to α-Btx-induced lethality.

## Glycosylation sites of nAChR subunits by α-Btx binding

We next examined glycosylation sites on nAChR subunits since these are known to have an important role in α-Btx binding affinity in other systems, and the identification of glycans on receptor subunits may help in the development of more species-specific insecticides (*Knight et al., 2004*). For example, deglycosylation reduces α-Btx binding in human nAChRs by more than two orders of magnitude (*Dellisanti et al., 2007*) and α-Btx binding to loop C in *Torpedo californica* α-subunits is enhanced by N-glycosylation of sites in these regions (*Rahman et al., 2020*). To identify specific glycosylation sites in *D. melanogaster* nAChRs we first purified SMALP solubilised receptors with α-Btx affinity beads, digested them into peptides and enriched for glycopeptides using HILIC resin (*Hägglund et al., 2004*, *Figure 6A*).

Site-specific identification of glycans on peptides by mass spectrometry is challenging (*Fang et al., 2020*) and often requires an additional deglycosylation step for glycopeptide measurement. Deglycosylation of enriched peptides was carried out using two separate enzymes: Endoglycosidase H (Endo H), which cleaves asparagine-linked oligosaccharides to generate a truncated sugar molecule with one N-acetylhexosamine (HexNAc) residue, and the endoglycosidase PNGase F, which releases the entire glycan from asparagine residues and deaminates the sugar free asparagine to aspartic acid. While very few glycopeptides were observed in the flow through (an average 20 glycopeptides, *Figure 6B*), we identified a total of 397 glycopeptides after enrichment and deglycosylation with Endo H or PNGase F (*Figure 6C*).

Shared glycopeptides from Dα5 and Dα7 nAChR subunits were identified after enrichment and deglycosylation with Endo H or PNGase F (*Figure 6D*). Deglycosylation with Endo H identified modified asparagine (N2) residues on the peptide (NNGSCLYVPPGIFK), which is predicted to be part of the Dα5 and Dα7 ligand-binding domains involved in α-Btx binding. This asparagine residue was modified with an N-acetylhexosamine (HexNAc) truncated sugar chain. Releasing N-glycans after

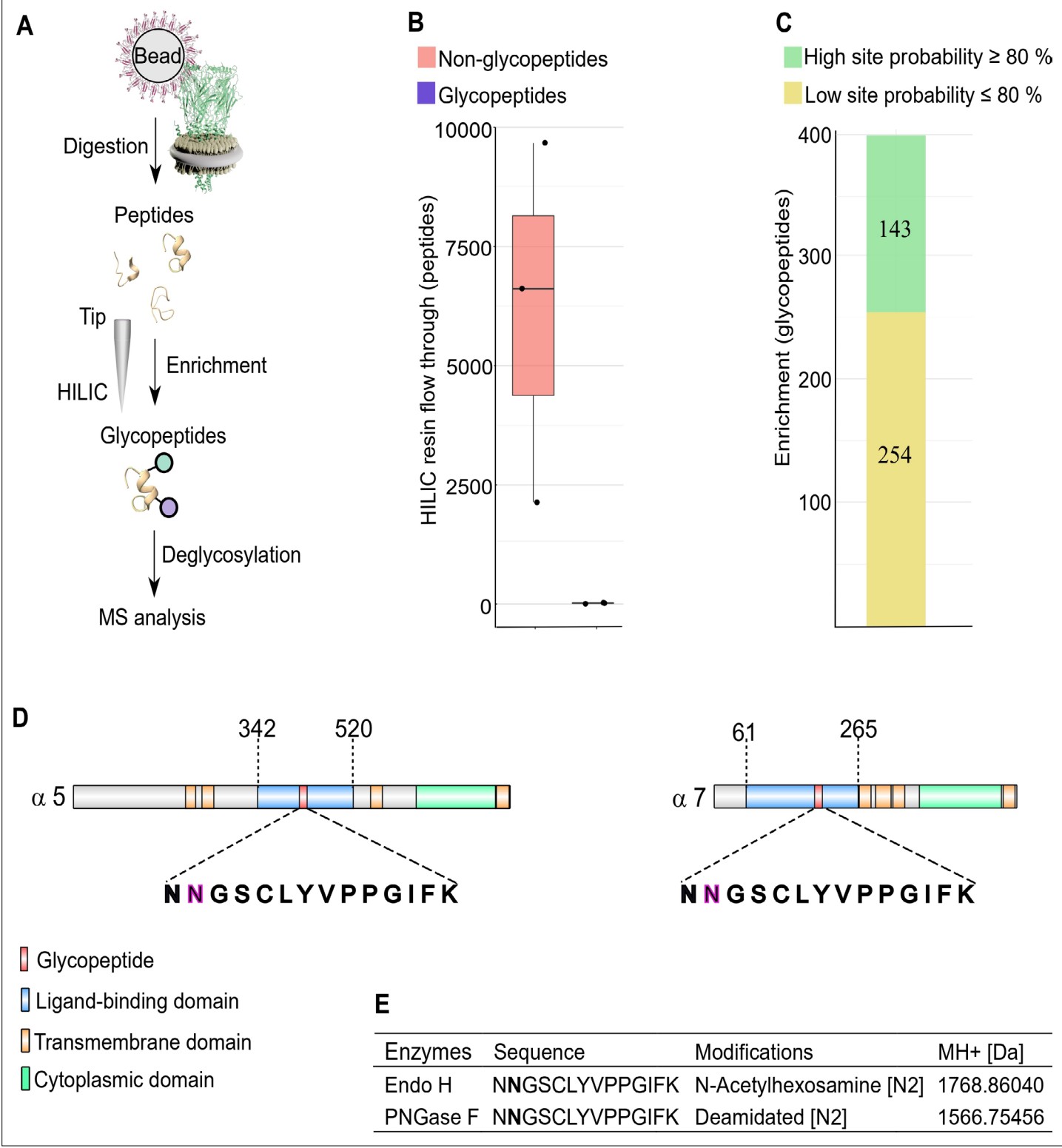

**Figure 6.** N-glycosylation sites in nAChR subunits. (**A**) Diagrammatic representation of nAChR subunit glycopeptide enrichment. Pull-downs with α-Btx affinity beads enrich for nAChRs and after tryptic digestion glycopeptides were enriched. Glycopeptides were deglycosylated with Endo H or PNGase F and analyzed by mass spectrometry. (**B**) Low numbers of glycopeptides (average 20) are detected in HILIC resin flow through fractions. (**C**) Numbers of identified glycopeptides according to site probabilities are shown (n=3). (**D**) Shared glycopeptide identified in the ligand-binding domain of Dα5 and Dα7, an N-linked glycosylated asparagine (N) residue is highlighted. (**E**) Deglycosylated peptide with either Endo H or PNGase F and contains either an

*Figure 6 continued on next page*

*Figure 6 continued*

N-acetylhexosamine or is deamidated on asparagine (N2). The two different modifications on the same peptide lead to a different monoisotopic mass (MH+ [Da]). Peptide contains an additional carbamidomethyl on cysteine (C5).

The online version of this article includes the following source data and figure supplement(s) for figure 6:

**Source data 1.** Glycopeptide raw data.

**Figure supplement 1.** MS/MS spectra of Dα5 and Dα7 subunit peptides.

**Figure supplement 2.** Glycosylation sites of nAChR subunits.

deglycosylation by PNGase F enabled us to identify a deaminated asparagine residue in the same peptide. The monoisotopic mass of this peptide changed due to this modifications on the second asparagine residue of this peptide (*Figure 6E*, *Figure 6—figure supplement 1*).

The genome of *Caenorhabditis elegans* encodes for at least 29 nAChR subunits (*Jones et al., 2007*) and the alpha-type unc-63 subunit contains an N-linked HexNAc modified asparagine residue on position 136 (*Kaji et al., 2007*). Performing a multiple sequence alignment showed that this asparagine residue is conserved between insects and nematodes (*Figure 6—figure supplement 2A*). Comparing identified glycosylation sites of Dα5 and Dα7 subunits to known N-linked glycosylation sites of α-subunits from *T. californica*, *Danio rerio*, *Mus musculus*, or *Homo sapiens* indicates that this site does not appear to be conserved between vertebrates and invertebrates (*Figure 6—figure supplement 2B*).

We also identified glycosylation sites in the Dα3 (ATKATLNYTGR) and Dβ3 (VVLPENGTAR) subunits after Endo H treatment but not with PNGase F treatment, suggesting they harbour a single N-linked HexNAc-modified asparagine residue (*Figure 6—figure supplement 2C*).

Taken together, these findings suggest that the Dα5 and Dα7 subunits are modified at asparagine residues in the α-Btx ligand-binding domain with an N-linked sugar chain.

## Localisation of Dα6 nAChRs subunit in the brain

The in vivo distribution of receptor subunits can provide important clues to understanding their biological function. While localisation studies with fluorescent protein tagged Dα6 subunits have been reported (*Perry et al., 2015*; *Martelli et al., 2022*) these have used transgenic constructs driven by Gal4 lines under the control of *nAChRα6* regulatory sequences. To characterise the endogenous localisation of an α-Btx binding receptor subunit, we elected to use CRISPR/Cas9 genome engineering to introduce in frame C-terminal fluorescence and epitope tags into the *nAChRα6* locus (*Figure 7*). We elected to use Dα6 in this exploratory study since there is localisation data from the studies mentioned above, as well as recent transcript localisation data (*Mitchell et al., 2021*), that we could use to validate out tagging strategy.

The resulting line was homozygous viable, fertile and showed no apparent phenotypes: we particularly noted that the curled abdomen phenotype observed at low frequency in the null allele was not detected in the tagged line. We used live confocal imaging with unfixed brains from larvae and adults homozygous for the tagged line. In 2nd instar larvae, we observed low level well-distributed fluorescence signal throughout the ventral nerve cord (VNC), including on commissural axons, and in the developing brain (*Figure 7A*).

By early 3rd instar larvae, we found more defined localisation in the VNC and developing mushroom bodies (*Figure 7B*), particularly noticeable in the Kenyon cells, a known site of α-Btx binding (*Su and O'Dowd, 2003*). Localisation in larval mushroom bodies continued to evolve in late 3rd instar larvae, with defined expression in the Kenyon cells, calyx, peduncle, dorsal and medial lobes (*Figure 7C*; *Video 1*), and with the strong mushroom body and VNC signal also observed when we imaged YFP in fixed preparations (*Figure 7D*). We also detected weaker signals in the medulla and lamina of the emerging optic lobes (*Figure 7E*; *Video 2*, *Figure 7F*; *Video 3*) as well as localisation to a number of cell bodies overlying the optic lobes (*Figure 7G*).

Finally, in the adult brain, expression was largely restricted to the mushroom bodies particularly the Kenyon cells and connections across the midline between the β and γ lobes and the optic lobes (*Figure 7H*; *Video 4*). The temporal localisation of Dα6 subunit in the CNS is summarised in schematic form (*Figure 7I*).

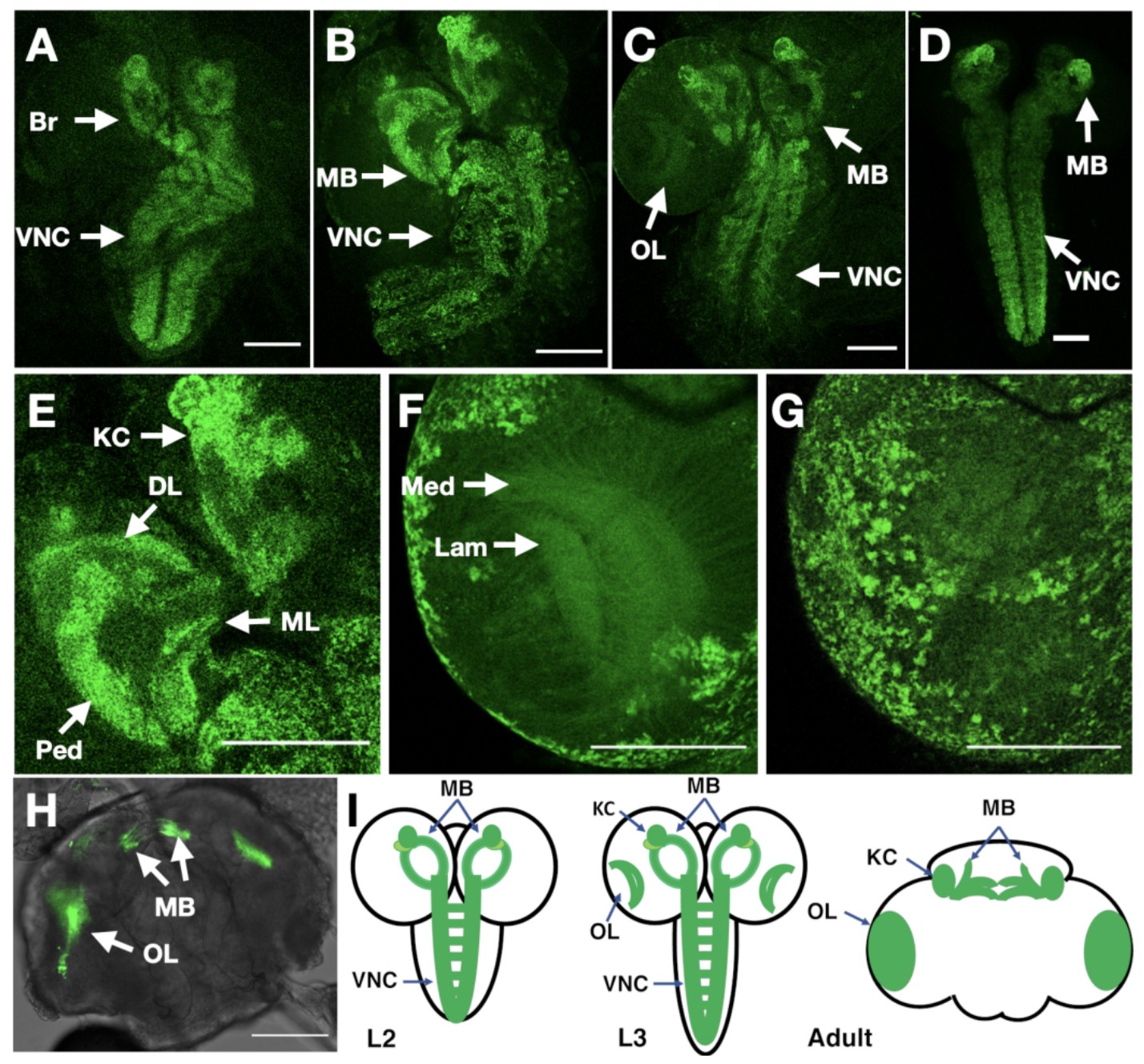

**Figure 7.** In vivo imaging of endogenously tagged Dα6 nAChR subunit. (**A–C**) Dα6-YFP localisation from live confocal imaging in 2nd A, early B and late C 3rd instar larval brains. Visible localisation in brain (Br), ventral nerve cord (VNC), mushroom bodies (MB), and optic lobes (OL). Scale bar = 100 μm. (**D**) Imaging Dα6-YFP localisation in fixed late 3rd instar larval brain. (**E**) Dα6-YFP in mushroom bodies of 3rd instar larvae with detectable fluorescence signal in Kenyon cells (KC), calyx (CX), peduncle (Ped), dorsal lobes (DL), and medial lobes (ML). Scale bar = 100 μm. (**F**) Dα6-YFP was observed in developing optic lobes, lamina (Lam) and medulla (Med) of later 3rd instar larvae. Scale bar = 100 μm. (**G**) Dα6-YFP on external structures of developing lobes in later 3rd instar larvae. Scale bar = 100 μm. (**H**) Dα6-YFP in the adult fly brain, strong signal is detected in mushroom bodies (MB) and optic lobe (OL). Scale bar = 100 μm. (**I**) Schematic summary of Dα6 subunit expression during different developmental stages, (L2, L3 and Adult) in which the green shading indicate the localisation of the Dα6 subunit.

The online version of this article includes the following figure supplement(s) for figure 7:

**Figure supplement 1.** In vivo imaging of endogenously tagged Dα1, Dβ1, Dβ2 nAChR subunits.

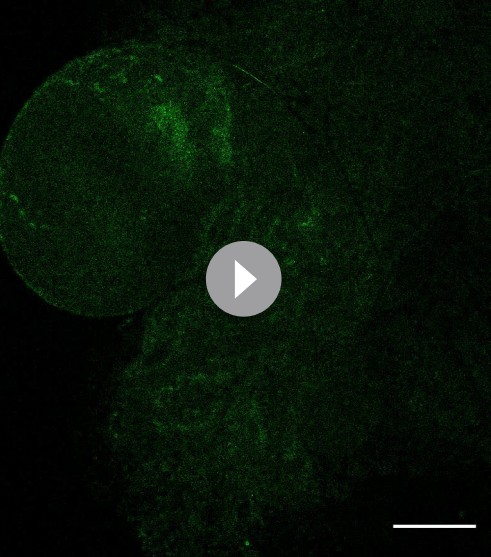

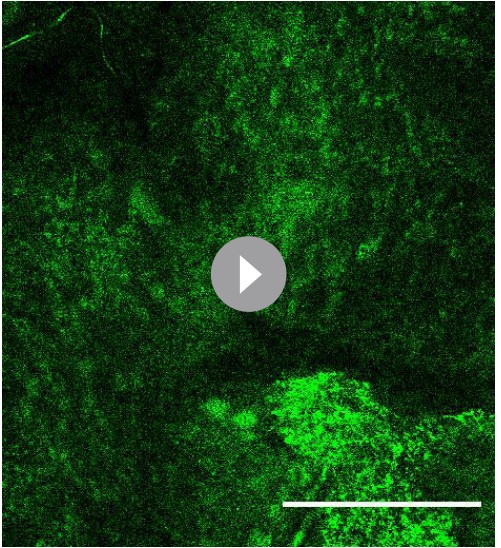

**Video 1.** Late 3rd instar larvae, with defined expression of Dα6-YFP in the Kenyon cells, calyx, peduncle, dorsal and medial lobes. Scale bar = 100 μm.

https://elifesciences.org/articles/74322/figures#video1

**Video 2.** Third instar larvae, with detectable fluorescence signal in Kenyon cells, calyx, peduncle, dorsal lobes and medial lobes. Scale bar = 100 μm.

https://elifesciences.org/articles/74322/figures#video2

The imaging of endogenously tagged Dα6 is largely consistent with the results reported from the expression of tagged transgenic lines (*Perry et al., 2015*; *Martelli et al., 2022*) and reported transcript localisation in mushroom bodies (*Mitchell et al., 2021*) with the main difference being the weak optic lobe expression we detected in larval brains. We detected strong optic lobe expression in the adult brain, consistent with previous reports (*Martelli et al., 2022*) and consider that the earlier expression detected with the endogenous fusion compared to Gal4 driven constructs likely reflects the regulatory complexity of the locus not captured by Gal4 drivers.

Taken together, our analysis indicates that, at least for Dα6, C-terminal tagging at the endogenous locus provides a faithful reporter of protein localisation and coupled with our α-Btx purification results presented above (*Figure 3C*), indicate that the tagged subunit is functional, at least in terms of toxin binding. We generated preliminary data tagging Dα1, Dβ1, and Dβ2 (*Figure 7—figure supplement 1*) using the same strategy and while

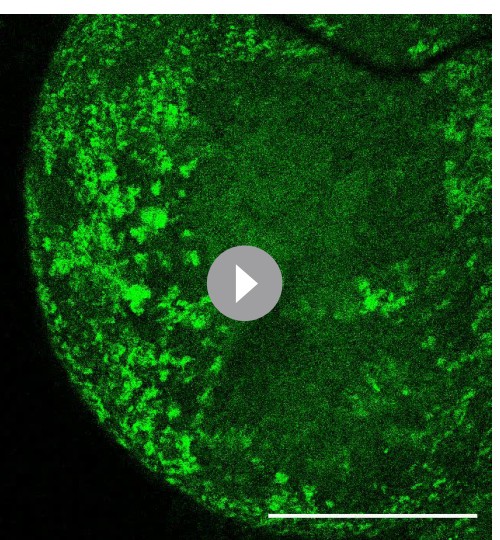

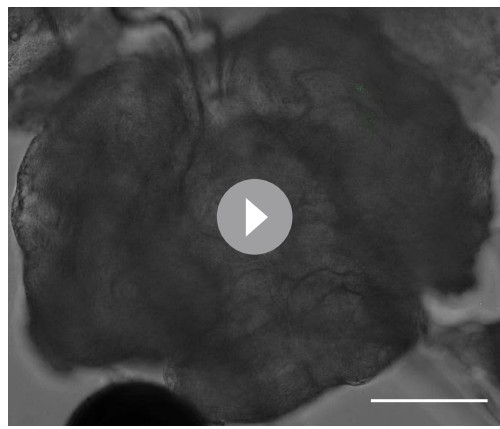

**Video 3.** Later 3rd instar larvae, with signal of Dα6-YFP in developing optic lobes, lamina (Lam) and medulla (Med). Scale bar = 100 μm.

https://elifesciences.org/articles/74322/figures#video3

**Video 4.** Adult brain, expression of Da6-YFP is largely restricted to the mushroom bodies. Scale bar = 100 μm.

https://elifesciences.org/articles/74322/figures#video4

we have not analysed these lines in detail it does suggest our endogenous tagging strategy may be of broad applicability.

## Discussion

Elucidation of complex insect nAChR heterogeneity will lead to a better understanding of selective insecticidal effects. We present a new set of null mutations in all *D. melanogaster nAChR* subunit genes and investigated insecticidal peptide toxin effects on wild-type and receptor subunit mutant larvae. Utilising biochemical approaches involving SMALP solubilisation to create nano-discs and pull-downs we characterised toxin binding of native nAChR complexes.

Our genome engineering approach generated viable and fertile mutations in nine out of the ten subunit genes encoded in the *D. melanogaster* genome and is largely concordant with the described work by Perry and colleagues (*Perry et al., 2021*). In both studies, null mutations in the nAChRβ1 gene were inviable as stocks. However, Christesen and colleagues report lethality primarily at the pre-pupal/pupal stage with a low percentage of individuals surviving to adulthood (*Christesen et al., 2021*). In contrast, we never recovered homozygous adults or pupae and when outcrossing to a 3rd chromosome balancer with a dominant larval marker did not find homozygous *nAChRβ1* 3rd instar larvae. It is possible that our DsRED insertion strategy somehow generated a stronger or neomorphic allele of *nAChRβ1*.

It is also possible that the lethality is due to a second site mutation on the chromosome or that there is a genetic background effect. We have also added to previous work by generating a new null mutation in *nAChRα5*, which shows resistance to α-Btx and a moderate reduction in locomotor activity. We observed some minor morphological defects in some of the null mutants especially in *nAChRα1, nAChRα2, nAChRα5* and *nAChRβ3* as well as locomotor defects with some alleles, particularly severely in *nAChRα3*. The locomotor defects we observed are in agreement with previously reported neuronal phenotypes with *nAChR* subunit genes, including sleep disruption, defective jump response, memory impairment or locomotor defects (*Fayyazuddin et al., 2006*; *Rohde et al., 2016*; *Somers et al., 2017*; *Tackenberg et al., 2020*). Locomotor defects in climbing assays were previously shown in *nAChRβ1* mutant strains (*Homem et al., 2020*; *Christesen et al., 2021*).

We used the nAChR null mutants to study insecticidal effects of the Hv1a peptide on viability after injection into larvae and investigated whether α-Btx has any insecticidal properties. As described by Chambers and colleagues, we confirm that Hv1a effects nAChRs (*Chambers et al., 2019*) and our analysis shows that the Dα4 and Dβ2 subunits are involved in the insecticidal response. We show for the first time that α-Btx has selective insecticidal effects against the Dα5, Dα6, and Dα7 subunits, which we further characterised at a biochemical level.

The pharmacology of Hv1a and α-Btx binding has been shown to be distinctive (*Chambers et al., 2019*), correlating with our demonstration that these two peptide toxins mediate their effects through different receptor subunits. Furthermore, resistance to neonicotinoid insecticides, which interact most strongly with Hv1a binding, has been associated with Dβ2 (*Perry et al., 2008*; *Perry et al., 2021*), consistent with the involvement of this subunit in the response to Hv1a. However, no resistance to neonicotinoids was seen in *D. melanogaster* carrying a *nAChRα4* gene knockout (*Perry et al., 2021*), which could be explained if neonicotinoids act on multiple receptor classes. Multiple binding sites for the neonicotinoid imidacloprid can be resolved in equilibrium binding assays in many insect species (*Xu et al., 2010*) and by binding kinetics in flies (*Liu and Casida, 1993*).

The agonist actions of neonicotinoids were investigated with *D. melanogaster* Dα1/Dβ1, Dα1/Dα2/Dβ1, Dα1/Dβ1/Dβ2, and Dα1/Dα2/Dβ1/Dβ2 nAChRs expressed in *Xenopus laevis* oocytes, indicating that Dβ1 subunit is a key player in the recognition of these insecticides (*Ihara et al., 2020*). It would be interesting to know if Dβ1 also plays a role in the in vivo interaction with insecticidal peptide toxins such as α-Btx or Hv1a, which is difficult to fully investigate because of the deleterious effects of *nAChRβ1* mutations. However, we note that while multiple Dβ1 peptides were recovered in our SMALP preparations without α-Btx enrichment, none were observed in the α-Btx pull-down experiments, suggesting that Dβ1 is not a major component of α-Btx binding receptors.

Resistance to spinosad is strongly associated with Dα6 (*Perry et al., 2021*), and spinosad binding is much more sensitive to the action of α-Btx than to the action of neonicotinoids (*Chambers et al., 2019*), again consistent with the involvement of this subunit with sensitivity to injected α-Btx and with the proposition that α-Btx and Hv1a act at distinct receptor classes. nAChR subunits are known to

be difficult to purify due to solubilisation issues (*Cheng et al., 2015*; *Maldonado-Hernández et al., 2020*) and the requirement for a lipid environment for ligand-binding (*daCosta et al., 2013*) makes it challenging to study these receptors in native conditions.

We used the SMALPs extraction method for preparing membrane discs and enriched nAChRs via α-Btx affinity purification and our Western blot analysis showed that Dα6-mVenus is enriched by α-Btx resins. Electron microscopy analysis indicated that receptor-like particles were recovered and these were substantially enriched by α-Btx pull-down. Mass spectrometry analysis showed an enrichment for the Dα5, Dα6 and Dα7 subunits in these preparations, which is concordant with our in vivo injection results and previous studies that characterised aspects of α-Btx binding (*Lansdell and Millar, 2004*; *Wu et al., 2005*; *Lansdell et al., 2012*). This confirms that in vivo molecular interactions are similar to those described in the S2 cell expression system.

To our knowledge this is the first report of the identification of a native endogenous α-Btx binding nAChRs. We note however, that we cannot determine from our analysis whether all three identified subunits are part of the same complex or if there are different receptors containing a subset of these subunits.

Using chimeric receptors in a cell line system, Landsdell and colleagues reported that a combination of all three of these subunits show high-affinity acetylcholine binding but α-Btx binding varied depending on receptor combinations, with Dα5 and Dα6 binding most strongly (*Lansdell et al., 2012*). In a prior study, they implicated Dα6 and Dα7 (*Lansdell and Millar, 2004*). However, these assays were performed with 5HT3A-nAChR subunit fusions, here we provide strong evidence that these three subunits bind to α-Btx in vitro and in vivo.

We note that at least some of the α-Btx binding receptors are likely to contain at least two Dα6 subunits since our non-reducing Western blots showed the presence of a dimer in the pull-downs. In addition, glycopeptide enrichment showed site specific glycosylation modifications on the Dα5 and Dα7 nAChR subunit ligand-binding domains. The unique lipid environment and glycosylation sites of nAChR α-subunits from the electric ray, *T. californica*, were found to be important for α-Btx binding activities (*Quesada et al., 2016*; *Rahman et al., 2020*), and structural studies support this conclusion (*Dellisanti et al., 2007*).

Our work supports the view that there is a role for Dα5 and Dα7 glycosylation modifications in the recognition of α-Btx in *D. melanogaster*. We note that Lansdell and colleagues have shown that Dα3 is implicated in α-Btx binding if it is co-expressed together with a non α-subunit such as mammalian b2, γ or δ in S2 cells (*Lansdell and Millar, 2000*). We did observe single peptides from Dα3 and Dβ3 in our α-Btx enrichments and our glycoproteomic analysis with α-Btx binding pull-downs identified additional shared peptides from Dα5 and Dα7 but also Dα3 and Dβ3, suggesting that there may be at least some α-Btx binding to receptors containing Dα3 together with a non α-subunit like Dβ3.

Localisation studies with fluorescence tagged endogenous Dα6 subunit, along with preliminary data from further three subunits indicates that a strategy generating C-terminal tagged subunits may be effective in following endogenous nAChR expression and opens the possibility of exploring receptor subunit composition using subunits with different fluorescent tags. Our analysis of Dα6 localisation showed relatively restricted expression in the brain and ventral nerve cord, with prominent expression in the Kenyon cells of the mushroom body. Our observations are largely consistent with recent reports of *nAChRα6* expression derived from expression reporters, particularly the strong mushroom body expression (*Perry et al., 2015*; *Martelli et al., 2022*), although we do observe weaker expression in the larval optic lobes, prefacing the later expression in the adult brain.

Work generating in locus Gal4 fusions for a variety of neurotransmitter receptors (*Kondo et al., 2020*) would seem to indicate wider adult brain expression than we and others observed, perhaps reflecting a degree of translational control or, more likely, limitations in the sensitivity of our live imaging compared to Gal4 driven reporters. The expression of Dα6 in Kenyon cells across development is in line with a proposed role for this subunit in memory plasticity, along with other α-subunits including Dα5, in mushroom body output neurons (*Barnstedt et al., 2016*). While our *nAChRα6* mutant shows significant resistance to α-Btx we still find α-Btx binding in the absence of Dα6 in our pull-down assays, likely a reflection of its restricted localisation. In contrast, it is clear that Dα6 plays a major and specific role in binding to the insecticide spinosad in *D. melanogaster* since mutations in this subunit are highly resistant to the toxin (*Perry et al., 2015*).

Translating our neurotoxin binding results to other target species, there are several studies showing that nAChRs are sensitive to α-Btx in other insects, including the cockroach *Periplaneta americana*, the moth *Manduca sexta* and the housefly *Musca domestica* (*Salgado and Saar, 2004*; *Eastham et al., 1998*; *Jones et al., 1981*). The exact α-Btx binding mechanism in these organisms is unknown or under investigation (*Salgado, 2021*). This indicates broader implications for our in vivo study determining the α-Btx binding nAChR subunits in the *D. melanogaster*, with our results potentially extrapolated to other target species that have similar nAChR subunits.

In conclusion, we identified ligand-binding subunit sites for a *D. melanogaster* nAChR antagonist with newly insecticidal effects. Our findings contribute to a better understanding of the role of nAChR subunits that interacts with insecticidal peptide toxins.

# Materials and methods

## Key resources table

| Reagent type (species) or resource | Designation | Source or reference | Identifiers | Additional information |
|---|---|---|---|---|
| Genetic reagent | pCDF3 | *Port et al., 2014* | N/A | |
| Genetic reagent | FSVS | *Korona et al., 2020* | N/A | |
| Genetic reagent | 3 × P3-DsRED | *Gratz et al., 2014* | N/A | |
| Genetic reagent (*D. melanogaster*) | {nos-Cas9}attP40 | *Ren et al., 2013* | BDSC:78,781 | |
| Genetic reagent (*D. melanogaster*) | {nos-Cas9}attP2 | *Ren et al., 2013* | BDSC:78,782 | |
| Genetic reagent (*D. melanogaster*) | w[1118] | FlyBase | FBal0018186 | |
| Genetic reagent (*D. melanogaster*) | nAChRα1 KO | This study | N/A | Steven Russell/ https://www.flyfacility.gen.cam.ac.uk |
| Genetic reagent (*D. melanogaster*) | nAChRα2 KO | This study | N/A | Steven Russell/ https://www.flyfacility.gen.cam.ac.uk |
| Genetic reagent (*D. melanogaster*) | nAChRα3 KO | This study | N/A | Steven Russell/ https://www.flyfacility.gen.cam.ac.uk |
| Genetic reagent (*D. melanogaster*) | nAChRα4 KO | This study | N/A | Steven Russell/ https://www.flyfacility.gen.cam.ac.uk |
| Genetic reagent (*D. melanogaster*) | nAChRα5 KO | This study | N/A | Steven Russell/ https://www.flyfacility.gen.cam.ac.uk |
| Genetic reagent (*D. melanogaster*) | nAChRα6 KO | This study | N/A | Steven Russell/ https://www.flyfacility.gen.cam.ac.uk |
| Genetic reagent (*D. melanogaster*) | nAChRα7 KO | This study | N/A | Steven Russell/ https://www.flyfacility.gen.cam.ac.uk |
| Genetic reagent (*D. melanogaster*) | nAChRβ2 KO | This study | N/A | Steven Russell/ https://www.flyfacility.gen.cam.ac.uk |
| Genetic reagent (*D. melanogaster*) | nAChRβ3 KO | This study | N/A | Steven Russell/ https://www.flyfacility.gen.cam.ac.uk |
| Genetic reagent (*D. melanogaster*) | nAChRα1[FSVS] | This study | N/A | Steven Russell/ https://www.flyfacility.gen.cam.ac.uk |
| Genetic reagent (*D. melanogaster*) | nAChRα6[FSVS] | This study | N/A | Steven Russell/ https://www.flyfacility.gen.cam.ac.uk |
| Genetic reagent (*D. melanogaster*) | nAChRβ1[FSVS] | This study | N/A | Steven Russell/ https://www.flyfacility.gen.cam.ac.uk |
| Genetic reagent (*D. melanogaster*) | nAChRβ2[FSVS] | This study | N/A | Steven Russell/ https://www.flyfacility.gen.cam.ac.uk |
| Genetic reagent (*E. coli*) | *E. coli* competent cells | New England Biolabs | Catalogue number: C2987H | |
| Commercial assay or kit | *Bbs*I | New England Biolabs | Catalogue number: R0539 | |
| Commercial assay or kit | Gibson Assembly Master Mix | New England Biolabs | Catalogue number: E2611L | |

*Continued on next page*

*Continued*

| Reagent type (species) or resource | Designation | Source or reference | Identifiers | Additional information |
|---|---|---|---|---|
| Commercial assay or kit | Q5 High-Fidelity 2 X Master Mix | New England Biolabs | Catalogue number: M0492L | |
| Commercial assay or kit | Pierce fluorometric peptide kit | Thermo Scientific | Catalogue number: 23,290 | |
| Peptide, recombinant protein | Trypsin/Lys-C mix | Promega | Catalogue number: V5073 | |
| Antibody | Anti-GFP (goat monoclonal) | Abcam | Catalogue number: Ab252881 | 1:1,000 |
| Antibody | Anti-rat IgG (goat polyclonal) | Sigma-Aldrich | Catalogue number: A9037 | 1:10.000 |
| Peptide, recombinant protein | $\omega$-hexatoxin-Hv1a | Syngenta | | CH-4332 Stein, Switzerland |
| Peptide, recombinant protein | α-Bungarotoxin | Abcam | Catalogue number: ab120542 | |
| Peptide, recombinant protein | Carbachol | Insight Biotechnology Ltd | Catalogue number: CAS 51-83-2 | |
| Other | CNBr-activated sepharose | Sigma-Aldrich | Catalogue number: C9 142–5 G | |
| Other | Styrene maleic acid copolymer (3:1) | Dafforn, p.c. | | |
| Commercial assay or kit | Mini-Protean TGX precast gels | Bio-Rad Laboratories | Catalogue number: 456–1,084 | |
| Commercial assay or kit | ECL-Chemiluminescent detection solution | GE Healthcare | Catalogue number: 45-000-999 | |
| Commercial assay or kit | CL-XPosure films | Thermo Scientific | Catalogue number: 10465145 | |
| Software, algorithm | GraphPad Prism | GraphPad Software | RRID:SCR_002798 | |
| Software, algorithm | Proteome Discoverer 2.3 | Thermo Scientific | RRID:SCR_014477 | |

## *Drosophila* methods

Embryos from the *THattP40* (*y¹ sc v¹ sev²¹; P{y⁺ᵗ⁷·⁷ v⁺ᵗ¹·⁸ nos-Cas9.R}attP40*) or *THattP2* (*y¹ sc v¹ sev²¹; P{y⁺ᵗ⁷·⁷ v⁺ᵗ¹·⁸ nos-Cas9.R}attP2*) lines expressing *nos*-Cas9 were injected using standard procedures (Bloomington *Drosophila* Stock Centre). Donor DNA (500 ng/µL) in sterile H₂O was injected together with gRNA plasmids (100 ng/µL) as described previously (*Korona et al., 2020*). Individually selected surviving adults were crossed to *w¹¹¹⁸* and the progeny screened for DsRED fluorescence localised mostly to the eyes of transgenic flies: positive flies were balanced and homozygous stocks established where possible. The correct localisation of the insert was confirmed via PCR and sequencing. Transgenic flies were assessed for the phenotype using bright field microscope. For tagging of Dα6 and other nAChR subunits, the stocks were additionally subjected to Cre-recombination for marker removal and several independent lines were verified by PCR. Some of these lines were screened for YFP fluorescence using confocal microscopy. From the YFP-positive balanced stocks, a viable and fertile homozygote stock was established. Injections were performed by the Department of Genetics Fly Facility (https://www.flyfacility.gen.cam.ac.uk). All fly stocks were maintained at 25°C on standard cornmeal medium.

## Cloning of gRNAs and generation of donor vectors

### Construction of *nAChR* subunits null alleles

In order to generate individual *nAChR* subunits gene mutations, the open-reading frame (ORF) was disrupted by introducing a visible marker harbouring DsRED marker under the eye specific 3P × 3 promoter using CRISPR/Cas9 technology as previously described (*Korona et al., 2020*). The targeted exons are shared between different isoforms and adjacent to the N-terminus to ensure the protein translation was interrupted. The insertion sites were designed in silico and optimal gRNAs were chosen (*Supplementary file 6*) that were tested against the injection strain and cloned into pCDF3. Briefly, target-specific sequences were synthesised and either 5'-phosphorylated annealed and ligated into the *Bbs*I sites of pCDF3 pre-cut with *Bbs*I. Positive clones were confirmed by sequencing. For generation of donor vectors, firstly, homology arms were amplified on genomic DNA (*Supplementary*

*file 7*) that, secondly, were used as a template to amplify the homology arms (*Supplementary file 8*) of the donor vector for CRISPR/Cas9 homologous recombination (HDR). The inserts with visible marker were amplified using as a template previously generated constructs (*Korona et al., 2020*) with appropriate primers.

These fragments were used for Gibson Assembly using Gibson Assembly Master Mix (New England Biolabs). PCR products were produced with the Q5 High-Fidelity 2 X Master Mix (New England Biolabs). All inserts were verified by sequencing.

## C-terminal tagging of Dα6 nAChRs subunit fusion protein

For tagging of Dα6 nAChRs subunit, the C-terminal fusion with FSVS fluorescent protein harbouring StrepII and 3xFLAG epitope tags (3xFLAG-StrepII-mVenus-StrepII) was generated for CRISPR/Cas9 mediated genome engineering (*Korona et al., 2017*; *Korona et al., 2020*). First, gRNAs were designed (*Supplementary file 6*) and tested against the genomic DNA sequence of injection strains. The oligonucleotides were phosphorylated and ligated into *Bbs*I pre-cut pCDF3. The positive variants were confirmed by sequencing. The donor vector to generate protein fusion with fluorescent protein harbouring epitope tags was cloned in two steps strategy by initially creating a *nAChRα6-FSVS* donor and then adding the removable marker to generate a *nAChRα6-FSVS-loxP-3P × 3-DsRED-loxP* donor vector. The homology arms were amplified from genomic DNA (*Supplementary file 7*) and used to amplify homology arms for the *nAChRα6-FSVS* donor vector (*Supplementary file 8*), assembled using Gibson Assembly as described above. The *FSVS* tag was amplified from previously generated constructs (*Korona et al., 2017*) with appropriate overlapping oligonucleotides (*Supplementary file 8*). The construct was confirmed by Sanger sequencing and used as a template to generate donor vector with removable marker. The PCR fragments harbouring homology arms and *FSVS* tag were amplified from the *nAChRα6-FSVS* construct, whereas the *3P × 3-DsRED* with adjacent *loxP* sites was amplified using earlier generated constructs (*Korona et al., 2017*). The final donor vector was generated using Gibson Assembly as described above and positive variants were confirmed by sequencing. Other *nAChR* tagged lines were constructed as mentioned above.

## Confocal microscopy

Localisation of FSVS-tagged (3xFLAG-StrepII-mVenus-StrepII) Dα6 nAChRs subunit was visualised in dissected larvae brains via monitoring the YFP fluorescence (mVenus). Briefly, the larval brains were dissected and mounted in glycerol for live imaging. For fixed preparations 3rd instar larvae brains were dissected on ice in chilled PBS, fixed in 4% PFA in PBS for 20 min at room temperature, washed three times with PBS-T (phosphate-buffered saline – PBS + 0.2% Triton) and mounted in VECTASHIELD.

Images were acquired using a Leica SP8 confocal microscope or ZEISS LSM 880 confocal with appropriate spectral windows for mVenus, images were processed with Fiji or ImageJ.

## Locomotor behaviour

Adult female and male flies were collected shortly after eclosion and separated into 10 cohorts consisting of 10 flies (100 total) for each genotype. Flies were maintained at 25 °C and transferred to fresh food every three days. For the curled abdomen phenotype, 50 flies (25 males and 25 females) in three replicates were used to examine the phenotype, we did not differentiate between males and females, however, we noticed that the phenotype predominantly affected males. For the climbing assay, each cohort was transferred to a plugged 10 ml serological pipette, and allowed to acclimatize for 5 min. For each trial, flies were tapped down to the bottom of the vial, and the percentage of flies able to cross a 5 ml mark successfully within 10 s was recorded as the climbing index. Five trials were performed for each cohort, with a 1 min recovery period between each trial. Climbing assays were performed 10 days after eclosion.

## *Drosophila* larval injections

Injections were performed by using the Nanoliter 2000 (World Precision Instruments, Hertfordshire, United Kingdom) mounted on a micromanipulator (Narishige, London, United Kingdom). Micropipettes were pulled from glass capillary tubes (1.14 mm OD, 0.530 mm ±25 µm ID; #4878, WPI) using a laser-based micropipette puller (Sutter P-2000, Sutter Instrument, Novato, CA, USA). Third instar larvae (wandering stage) were transferred to an adhesive surface after being quickly

washed with water to remove food residues and gently dried using paper tissue. The micropipette was positioned over the approximate centre of the body, on the dorsal side, and the tip was advanced through the cuticle into the hemocoel of the larva. Larvae were injected with 69 nL of PBS (phosphate-buffered saline) supplemented with 10% (v/v) filtered food dye (PME, moss green food colouring; 0.2 µm filter). Food dye was included to aid in monitoring the success of the injection under a dissection microscope (Leica MZ65, Milton Keynes, United Kingdom). $\omega$-hexatoxin-Hv1a (Hv1a, Syngenta, Schaffhauserstrasse, CH-4332 Stein, Switzerland) and α-Bungarotoxin α-Btx (ab120542, Abcam, Cambridge, United Kingdom) were added to the injection mix in order to obtain a final concentration of 2.5 nmol/g and 1.25 nmol/g, respectively (average larval weight was 2.14 mg).

After injection, larvae were then gently transferred into agar/grape juice (Ritchie Products Limited, Burton-On-Trent, United Kingdom) plates and kept at 25 °C. The rate of survival (expressed as percentage) was calculated as the number of living pupae, formed 1–2 days after injection, divided by the total number of injected larvae. Experiments were repeated three times independently with a total number of 10 larvae for each experimental group. Results were analysed with One-way ANOVA followed by Bonferroni's multiple comparisons test using GraphPad Prism (version 7, GraphPad Software, San Diego, California, USA).

## Coupling procedure of α-Bungarotoxin to affinity beads

Coupling of α-Bungarotoxin, α-Btx (ab120542, Abcam, Cambridge, United Kingdom) to cyanogen bromide-activated (CNBr) sepharose beads 4B (C9 142–5 G, Sigma-Aldrich, Haverhill, United Kingdom) was performed as described (*Wang et al., 2003*; *Mulcahy et al., 2018*). CNBr-activated sepharose 4B beads (0.25 g) were hydrated in 1.25 ml of 1 mM HCl for 1 hr at 4 °C on a rotator. Beads were centrifuged for 5 min at 1500 × g, the supernatant removed and beads washed twice with 1 ml of coupling buffer (0.25 M NaHCO$_3$, 0.5 M NaCl, pH 8.3). Beads were centrifuged for 5 min at 1500 × g and the supernatant was removed. Alpha-Btx (1 mg) was resuspended in 1 ml coupling buffer and incubated together with the affinity beads at 4 °C for 16 hr on a rotator. Beads were centrifuged for 5 min at 1500 × g. Coupling efficiency was determined using a Pierce quantitative fluorometric peptide kit and used according to the manufacturer's instructions (23290, Thermo Scientific, Bishop's Stortford, United Kingdom). Beads were blocked with 1 ml of 0.2 M glycine in 80% coupling buffer at 4 °C for 16 hr on a rotator. Beads were then centrifuged for 5 min at 1500 × g and washed with 1 ml of 0.1 M NaHCO$_3$, 0.5 M NaCl, pH 8.0. This step was repeated with 1 ml of 0.1 M NaCH$_3$CO$_2$, 0.5 M NaCl, pH 4.0. Beads were washed again in 1 ml of 0.1 M NaHCO$_3$, 0.5 M NaCl, pH 8.0. After a final wash step with 1 ml coupling buffer the beads were incubated twice for 30 min in 1 ml Tris-buffer (50 mM Tris, 150 mM NaCl, pH 8.0). The beads were centrifuged for 5 min at 1500 × g, the supernatant was removed.

## Membrane protein enrichment and incorporation in SMALPs

*D. melanogaster* heads were obtained and separated according to *Depner et al., 2014*. In a 50 ml falcon tube approximately 6 g flies were rapidly frozen in liquid nitrogen and vortexed twice for 3 min, with the tube cooled for 30 sec in liquid nitrogen between. Heads were separated from bodies by sieving (1201124 & 1201125, Endecotts, London, United Kingdom).

1 ml of isotonic lysis buffer (0.25 M sucrose, 50 mM TRIS/HCl pH 7.4, 10 mM HEPES pH 7.4, 2 mM EDTA, Protease inhibitor) was added to approximately 0.8 g separated heads. The solution was mixed three times by vortexing and the heads were lysed with 60 strokes in a Dounce homogenizer with a pestle. Membrane protein preparation was performed by differential centrifugation-based fractionation as described (*Depner et al., 2014*; *Geladaki et al., 2019*). Membranes (24–177 mg wet pellet weight) were resuspended in approximately 20–300 µl 5% SMALP solution (5% styrene maleic acid copolymer (3:1), 5 mM Tris-Base, 0.15 mM NaCl, pH 8.0). As a comparison membrane pellets were also solubilised in 5% SMALP solution without the addition of the copolymer (*Supplementary file 9*). For efficient incorporation and formation of SMALPs, membrane proteins were incubated with 5% SMALP solution for 2 hr at room temperature on a rocking platform. To separate the insoluble proteins from the soluble SMALPs a centrifugation step at 100,000 × g for 60 min, 4 °C was performed. Supernatant containing the SMALPs was combined and used for the nAChRs pull-downs.

## Enrichment of nAChRs by α-Btx pull-down

SMALPs (20–35 mg/ml) were incubated with 200 µl α-Btx conjugated affinity beads for 16 hr, 4 °C on a rotator. The beads were then centrifuged for 5 min at 1500 × g and washed two or three times, each for 10 min with 1 ml ice-cold TBS (50 mM Tris, 150 mM NaCl, pH 8.0) on a rotator at 4 °C. Beads were centrifuged for 5 min at 1500 × g and nAChRs selectively eluted twice with 100 µl 1 M carbachol (CAS 51-83-2, Insight Biotechnology Ltd, Wembley, United Kingdom). These steps were performed for 25 min at room temperature on a rotator. Beads were centrifuged for 5 min at 1500 × g and eluates were combined and ice-cold 100% acetone in the volume of four times of the sample was added to the samples, mixed by vortexing and proteins were precipitated for 16 hr at –20 °C. Samples were centrifuged at 13000 × g for 15 min. Supernatant was removed and dried proteins were dissolved in Laemmli buffer (1 M Tris pH 6.8, 10% SDS, 5% glycerol, 2% bromophenol blue). Proteins were heated at 60 °C and loaded on Mini-Protean TGX precast gels (456–1084, 4–15 %, Bio-Rad Laboratories, Inc, Watford, United Kingdom).

## Immunoblotting

Alpha-Btx pull-down enriched or unenriched protein samples were treated with 1% DTT or left untreated, boiled at 60 °C for 8 min, separated by PAGE and transferred onto a nitrocellulose membrane (1704158, Trans-Blot Turbo Mini, Bio-Rad Laboratories, Inc, Watford, United Kingdom).

Ponceau S staining was used as a sample loading control. FSVS-tagged Dα6 (3xFLAG-StrepII-mVenus-StrepII) was detected with anti-GFP (Ab252881, Abcam, Cambridge, United Kingdom). 5% skimmed milk power dissolved in TBS-T was used for blocking and membranes were incubated for 16 hr at 4 °C with the α-GFP antibody (1:1000 concentrated in blocking solution) followed by anti-rat IgG antibody for 1 hour (A9037, Sigma-Aldrich, Haverhill, United Kingdom). Immunoblots were treated with an ECL chemiluminescent detection solution (45-000-999, GE Healthcare, Chalfont St. Giles, United Kingdom) exposed for 10 s to CL-XPosure films (10465145, Thermo Scientific, Bishop's Stortford, United Kingdom) and visualised using an x-ray developer (1170-1-8000, Protec GmbH, Oberstenfeld, Germany). Two biological replicates were performed.

## Electron microscopy preparation

For negative staining analysis, membrane proteins were extracted with 5% SMA and nAChRs were enriched using α-Btx affinity pull-downs. Proteins were diluted 1:10 with deionised water to approximately 0.9 mg/ml and an aliquot of the samples were absorbed onto a glow-discharged copper/carbon-film grid (EM Resolutions) for approximately 2 min at room temperature. Grids were rinsed twice in deionised water and negative staining was performed using a 2% aqueous uranyl acetate solution. Samples were viewed in a Tecnai G2 transmission electron microscope (TEM, FEI/ThermoFisher) run at 200 keV accelerating voltage using a 20 µm objective aperture to increase contrast; images were captured using an AMT CCD camera. Three biological replicates were performed and we provide 28 micrographs, 15 enriched from α-Btx and 13 unenriched (*Figure 3—source data 1*).

## Sample preparation for liquid chromatography–mass spectrometry (LC-MS)

Gel pieces were excised from the gel lanes and proteolytic digestion with Trypsin/Lys-C mix (V5073, Promega, Southampton, United Kingdom) was performed as described (*Shevchenko et al., 2006*). The gel pieces were covered with 50 mM NH₄HCO₃ / 50% ACN and shaken for 10 min. This step was repeated with 100% acetonitrile and finally dried in a speed vac. Samples were reduced with 10 mM DTT in 50 mM NH₄HCO₃ at 56 °C for 1 hr and alkylated with 50 mM iodoacetamide in 50 mM NH₄HCO₃ at room temperature without light for 45 min. The gels were covered with 50 mM NH₄HCO₃ and 100% ACN and shaken for 10 min. These steps were repeated and samples were dried in a speed vac. Trypsin/Lys-C buffer was added to the sample according to manufacturer's instructions and incubated for 45 min on ice.

Next 30 µl 25 mM NH₄HCO₃ was added and samples were incubated at 37 °C for 16 hr. The gel pieces were covered with 20 mM NH₄HCO₃ and shaken for 10 min. Supernatant with peptides was collected. Next, the gels were covered with 50% ACN / 5% FA and shaken for 20 min. These steps were repeated and peptides were dried in a speed vac. Samples for glycopeptide enrichment were digested in-solution according to *Queiroz et al., 2019*. Samples were reduced and alkylated

in 10 mM DTT and 50 mM iodoacetamide. Proteins were digested in final concentration of 2.5 µg Trypsin/Lys-C buffer for 16 hr at 37 °C.

## Peptide clean-ip

Peptides were desalted using C-18 stage tips according to *Rappsilber et al., 2007*. C-18 material (three C-18 plugs were pasted in a 200 µl pipette tip, Pierce C18 Spin Tips, 84,850 Thermo Scientific, Bishop's Stortford, United Kingdom) was equilibrated with methanol/ 0.1% FA, 70% ACN/ 0.1% FA and with 0.1% FA. Peptides were loaded on C-18 material, washed with 0.1% FA and eluted with 70% ACN/0.1% FA. Samples were dried and finally, peptides were resuspended in 20 µl 0.1% FA. For glycopeptide enrichment, peptides were first desalted using Poros oligo r3 resin (1-339-09, Thermo Scientific, Bishop's Stortford, United Kingdom) as described (*Gobom et al., 1999*; *Queiroz et al., 2019*). Pierce centrifuge columns (SH253723, Thermo Scientific, Bishop's Stortford, United Kingdom) were filed with 250 µl of Poros oligo r3 resin. Columns were washed three times with 0.1% TFA. Peptides were loaded onto the columns and washed three times with 0.1% TFA and subsequently eluted with 70% ACN.

## Glycopeptide enrichment

Enrichment of glycopeptides of nAChRs was performed as described (*Hägglund et al., 2004*). Micro columns were prepared with 200 µl peptide tips filled with a C8 plug and iHILIC – fusion 5 µm, 100 Å silica based material (HCS 160119, Hilicon, Umeå, Sweden). Peptides were solubilised stepwise in 19 µl dH$_2$O and then in 80 µl ACN plus 1 µl TFA acid. The micro columns were cleaned with 50 µl 0.1% TFA and three times equilibrated with 100 µl 80% ACN, 1% TFA. Peptides were loaded onto the micro column and washed twice with 100 µl 80% ACN, 1% TFA. Glycopeptides were eluted from the column using twice 40 µl 0.1% TFA and finally with 20 µl 80% ACN, 1% TFA. Samples were dried in a speed vac before peptides were deglycosylated with Endo H or PNGase F according to manufacturer's instructions (P07025 & P0710S, New England Biolabs Inc, Hitchin, United Kingdom).

## LC-MS/MS

Peptide samples were dissolved in 20 µl of 0.1% (v/v) FA. Approximately 1 µg peptide solution was used for each LC-MS/MS analysis. All LC-MS/MS experiments were performed using a Dionex Ultimate 3000 RSLC nanoUPLC (Thermo Fisher Scientific Inc, Waltham, MA, USA) system and a Q Exactive™ Orbitrap mass spectrometer (Thermo Fisher Scientific Inc, Waltham, MA, USA). Separation of peptides was performed by reverse-phase chromatography at a flow rate of 300 nL/min and a Thermo Scientific reverse-phase nano Easy-spray column (Thermo Scientific PepMap C18, 2 µm particle size, 100 Å pore size, 75 µm i.d. x 50 cm length). Peptides were loaded onto a pre-column (Thermo Scientific PepMap 100 C18, 5 µm particle size, 100 A pore size, 300 µm i.d. x 5 mm length) via the Ultimate 3,000 autosampler with 0.1% FA for 3 min at a flow rate of 15 µL/min. After loading, the column valve was switched to allow elution of peptides from the pre-column onto the analytical column. Solvent A was water +0.1% FA and solvent B was 80% ACN, 20% water +0.1% FA. The linear gradient employed was 2%–40% B in 90 min (the total run time including column washing and re-equilibration was 120 min). In between runs columns were washed at least four times to avoid carry overs. The LC eluant ionised by means of an Easy-spray source (Thermo Fisher Scientific Inc). An electrospray voltage of 2.1 kV was applied in order to ionise the eluant. All m/z values of eluting ions were measured in an Orbitrap mass analyzer, set at a resolution of 35,000 and scanned between m/z 380 and 1500. Data-dependent scans (Top 20) were employed to automatically isolate and generate fragment ions by higher energy collisional dissociation (HCD; Normalised collision energy (NCE): 25 %) in the HCD collision cell and measurement of the resulting fragment ions were performed in the Orbitrap analyser, set at a resolution of 17,500. Singly charged ions and ions with unassigned charge states were excluded from being selected for MS/MS and a dynamic exclusion of 20 s was employed.

## Peptide/protein database searching

Protein identification was carried out using Sequest HT or Mascot search engine software operating in Proteome Discoverer 2.3 (*Eng et al., 1994*; *Koenig et al., 2008*). Raw flies were searched against the Uniprot *Drosophila_melanogaster*_20180813 database (23,297 sequences; 16110808 residues) and a common contaminant sequences database. The search parameters using mascot algorithm were: (i)

Trypsin was set as the enzyme of choice, (ii) precursor ion mass tolerance 20 ppm, (iii) fragment ion mass tolerance 0.1 Da, (iv) maximum of two missed cleavage sites were set, (v) a minimum peptide length of six amino acids were set, (vi) fixed cysteine static modification by carbamidomethylation, (vii) variable modification by methionine oxidation & deamidation on asparagine and glutamine and N-acetylhexosamine (HexNAc(1)dHex(1) + HexNAc on asparagine) as variable glycopeptide modifications, (viii) A site probability threshold of 75% was set, (ix) Percolator was used to assess the false discovery rate and peptide filters were set to high confidence (FDR < 1).

## Data handling and statistical analysis

Protein data evaluation was performed using R 3.5.3 (*Ihaka and Gentleman, 1996*). Plotting of graphs were performed in RStudio 1.3.959 (*Rstudio Team, 2020*) using ggplot2 (*Ginestet, 2011*) and other R packages. In order to characterise membrane proteins, the following tools were used: (i) TMHMM - 2.0 (*Krogh et al., 2001*), (ii) PRED-TMBB2 (*Tsirigos et al., 2016*) (iii) SwissPalm (*Blanc et al., 2015*), (iv) PredGPI (*Pierleoni et al., 2008*), (v) Gravy calculator (https://www.gravy-calculator.de), (vi) Myristoylator (*Bologna et al., 2004*) (vii) Solubility scores (*Sormanni et al., 2015*; *Sormanni et al., 2017*). Analysis of gene ontology (GO) slim terms (*The Gene Ontology Consortium, 2019*) were performed within proteome discoverer 2.3 (Thermo Fisher Scientific). KEGG (*Kanehisa et al., 2020*) pathway enrichment analysis was performed using DAVID (*Huang et al., 2009*). For each experimental investigation, n ≥ 3 were considered and data are represented as means ± SEM. Experiments were performed in a blinded manner whenever possible. Data are presented as mean ± SD. Statistical tests for SMALPs were performed using two-tailed t-test with an unequal variance and p values of ≤ 0.05 were considered to be significant. In DAVID, Fisher's exact p values are computed to measure the gene-enrichment terms. Fisher's exact p value of 0 represents perfect enrichment of a term. Usually p value of ≤ 0.05 are to be considered as strongly enriched. In this study, the default threshold set in DAVID of 0.1 was used. Linear regression analysis was performed in order to study the efficiency of SMALPs extraction of membrane receptors.

## Structural assessment and illustration of nAChR subunits

For structural alignment of nAChRs matchmaker command operating in UCSF Chimera X 0.91 (*Goddard et al., 2018*) was used. This command is superimposing protein structures by first creating pairwise sequence alignments, then fitting the aligned residue pairs and displays in an overlaid structure as a result. The following parameters were set to create the aligned structure: (i) alignment algorithm; Needleman-Wunsch (ii) similarity matrix; BLOSUM-62. Structural animation was performed in Blender 2.8 (https://www.blender.org), an open-source 3D graphics software. For annotation of protein sequences InterProScan was used (*Mitchell et al., 2019*). Illustrator for biological sequences (IBS) web server was used to represent biological sequences (*Liu et al., 2015*). Multiple sequence alignments were performed (*Madeira et al., 2019*) or using BoxShade multiple sequence alignments (Swiss institute of bioinformatics).

# Acknowledgements

We thank Professor Tim Dafforn for kindly providing us styrene maleic acid (SMA) copolymer, Dr. Daniel Nightingale for helpful exchange and Mrs Renata Feret for technical discussions. We are very grateful to Syngenta, Milner Therapeutics Institute and the MRC Toxicology Unit for excellent infrastructure support. Electron microscopy was performed using the facilities at CAIC (Cambridge Advanced Imaging Centre, University of Cambridge). Funding was provided by BBSRC and Syngenta. For the purpose of Open Access, the authors have applied a CC BY public copyright licence to any Author Accepted Manuscript (AAM) version arising from this submission. Funding UKRI-BBSRC BB/P021107/1 Dagmara Korona, UKRI-BBSRC BB/P021107/1 Benedict Dirnberger. The funders had no role in study design, data collection and interpretation, or the decision to submit the work for publication.

## Additional information

### Competing interests

Benedict Dirnberger, Lucy C Firth: is affiliated with Syngenta. The author has no other competing interests to declare. The other authors declare that no competing interests exist.

### Funding

| Funder | Grant reference number | Author |
|---|---|---|
| Biotechnology and Biological Sciences Research Council | BB/P021107/1 | Dagmara Korona Benedict Dirnberger |

The funders had no role in study design, data collection and interpretation, or the decision to submit the work for publication.

### Author contributions

Dagmara Korona, Carlo NG Giachello, Formal analysis, Investigation, Validation, Visualization, Project administration, Resources, Supervision; Benedict Dirnberger, Formal analysis, Investigation, Validation, Visualization, Project administration, Resources, Supervision, Writing – review and editing; Rayner ML Queiroz, Formal analysis, Investigation, Visualization, Project administration; Rebeka Popovic, Formal analysis, Validation, Visualization, Project administration, Resources; Karin H Müller, Validation, Visualization, Resources; David-Paul Minde, Investigation, Visualization; Michael J Deery, Formal analysis, Visualization, Supervision; Glynnis Johnson, Visualization; Lucy C Firth, Writing – original draft; Fergus G Earley, Data curation, Formal analysis, Methodology, Validation, Visualization, Writing – original draft, Project administration, Resources, Supervision, Writing – review and editing; Steven Russell, Kathryn S Lilley, Conceptualization, Data curation, Formal analysis, Funding acquisition, Investigation, Methodology, Project administration, Resources, Supervision, Validation, Visualization, Writing – original draft, Writing – review and editing

### Author ORCIDs

Dagmara Korona ⬤ http://orcid.org/0000-0002-5988-3894
Benedict Dirnberger ⬤ http://orcid.org/0000-0002-0772-5923
Fergus G Earley ⬤ http://orcid.org/0000-0003-1943-0724

### Decision letter and Author response

Decision letter https://doi.org/10.7554/eLife.74322.sa1
Author response https://doi.org/10.7554/eLife.74322.sa2

## Additional files

### Supplementary files

• Supplementary file 1. Abdomen phenotype.

• Supplementary file 2. Climbing ability.

• Supplementary file 3. *Drosophila* larval injection of $\omega$-Hexatoxin-Hv1a & α-Bungarotoxin.

• Supplementary file 4. Identified nAChR peptides in pull-downs with α-Bungarotoxin. Peptides from Dα3, Dα5, Dα6, Dα7 and Dβ3 nAChR subunits are listed and found [N] times within individual replicates. Protein domains are marked with: Ed extracellular-, Id Intracellular-, LBD ligand-binding-, and Non-domain localization. The mass-to-charge ratio (m/z) of the precursor ions, the protonated monoisotopic masses, the theoretical MH$^+$ masses in Dalton [Da] and peptide modifications are listed. Peptide modifications are listed with: (C) Carbamidomethylation; (N,Q) Deamidation; (H) N-acetylhexosamine (HexNAc); (M) Oxidation.

• Supplementary file 5. Identified nAChR peptides in pull-downs without α-Bungarotoxin. Identified peptides of nAChR subunits which are found in control pull-down samples without α-Bungarotoxin (α-Btx).

• Supplementary file 6. List of gRNAs and oligonucleotides used for cloning.

• Supplementary file 7. List of oligonucleotides used for amplification from genomic DNA.

- Supplementary file 8. C-terminal tagging of nAChRa6 with FSVS.
- Supplementary file 9. Wet pellet weight of membrane fractions.
- Transparent reporting form

### Data availability

The mass spectrometry data from this publication have been deposited to PRIDE (http://www.ebi.ac.uk/pride/archive/) with the data set identifier PXD028484 (DOI: https://doi.org/10.6019/PXD028484).

The following dataset was generated:

| Author(s) | Year | Dataset title | Dataset URL | Database and Identifier |
|-----------|------|---------------|-------------|-------------------------|
| Benedict D | 2021 | Drosophila nicotinic acetylcholine receptor subunits and their native interactions with insecticidal peptide toxins | http://dx.doi.org/10.6019/PXD028484 | ProteomeXchange, 10.6019/PXD028484 |

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
