## [Editor Report]

The authors employ genetic and biochemical approaches to demonstrate the insecticidal effects of a snake peptide toxins. Intriguingly, they show that it targets different nicotinic acetylcholine receptor subunits than a previously identified insecticidal spider toxin. Especially their clever combination of detergent-free membrane protein extraction and mass spectrometry will no doubt prove useful to study native receptor-ligand interactions in the future.

---

## [Decision Letter]

**Decision letter after peer review:**

Thank you for submitting your article "*Drosophila* nicotinic acetylcholine receptor subunits and their native interactions with insecticidal peptide toxins" for consideration by *eLife*. Your article has been reviewed by 3 peer reviewers, and the evaluation has been overseen by a Reviewing Editor and K VijayRaghavan as the Senior Editor. The following individual involved in review of your submission has agreed to reveal their identity: Henriette Elisabeth Autzen (Reviewer #1).

Essential revisions:

1) Functional verification of the predicted subunit-specific toxin effects, also taking into account that these toxins bind at subunit interfaces.

2) Additional negative stain EM analysis to substantiate claims with adequate statistics, or adjust analysis and claims to reflect limitations.

3) Data on alpha5/7 localisation or a clear narrative and justification of why only alpha6 was subjected to localisation studies.

4) Language should be carefully revised based on the comments below to make it accessible to a broader audience. Additionally, the manuscript needs to better reflect prior work, provide greater clarity on the final genotypes of the knockout strains and provide additional context and motivation for the proteomic/glycosylation work.

A straightforward way to address these requests would be inclusion of additional results. Further details of these concerns are in the reviews, along with suggestions that we hope you will find helpful.

*Reviewer #1 (Recommendations for the authors):*

Further comments and suggestions to the authors:

1) One is left to wonder why the authors didn't run the solubilized sample on an SDS page gel and subject it to Western blotting with nAChR specific antibodies to show how α-Btx affinity resin helped enrich the sample with nAChR. This would have been more quantitative than their TEM analysis.

2) The authors should provide info on the amount of membrane (g) used in their SMA solubilization described in the methods section.

3) The authors should substantiate their TEM analysis and claims with statistics by collecting more than a single TEM micrograph and used TEM software to calculate 2D class averages and compare those to see if there are "side" and top views of nAChR. The authors should also provide details on how many images their analysis is based upon in the methods section.

4) Figure 3C which validates coupling of α-Btx to the resin rightfully belongs in a lab notebook, not as a figure in the main text.

5) The authors should include a description of how to calculate the hydrophobicity score.

6) Line 308. As a reader, one wonders why there is no evidence of peptides mapping to TMH domains. The authors should explain.

7) Lines 308-310. The authors should expand their description of the mapping of the identified peptides. It is not clear why the identified peptides are outside of the α-Btx binding sites with the information that is currently given.

8) A Figure like the one in Supplement 3B should be found in the main text figures and not in the supplement. However, the figure should be better annotated such that the reader can easily understand whether the α-Btx is docked into the model or part of it.

9) Figure 6B. It is not clear which flowthrough that is meant. From α-Btx resin or the HILIC resin.

10) Information on how the -SMALP sample was prepared is missing from the methods. Was it simply prepared without SMA in the buffer or was something different done?

*Reviewer #2 (Recommendations for the authors):*

This study was carefully and comprehensively conducted to demonstrate that the two peptide toxins differentially target insect nAChRs by in vivo and biochemical experiments. It is, however, noteworthy that the in vivo data may be the results of compensation by other nAChR subunits. In the paper the ventral nerve cords of adult flies were used to purify the proteins but when other preparations or parts are used, different results may be obtained. The pull-down assay identified the main targets, but in the case of α Btx, alpha3 was also identified but in much lower frequency, which should also be noted in the text as a fact. As regards localisation of the subunits, it is plausible to investigate the localisation of α5 and α7 subunits to examine if these subunits are colocalized with α6 to form heteromers as the toxin binding sites.

*Reviewer #3 (Recommendations for the authors):*

The use of the term deletion to describe the strains is not quite accurate. The genes are clearly disrupted by the constructs and so a more appropriate term along these lines could be considered.

For the injection assay results, presentation of the individual replicate data points on the figure 2 would be a clearer way to provide insights to the reader regarding variability rather than relying on the table. I also have a question about the description of the figure (line 190) of there being 10 larvae per replicate, however there are a number of data points where 25% is reported in Appendix Table 2. How was this calculated and would raw data be more suitable to report in the table given the number of larvae tested was 10 for each replicate?

Related to the point in the public review, if the Dbeta1 strain in this study is able to survive until pupation, would it be possible to conduct the injection experiments on these to assay for the involvement of Dbeta1, particularly in the action of Hvl1a? This would enable results from the full set of genes to be reported, strengthening the findings.

---

## [Author Response]

Essential revisions:1) Functional verification of the predicted subunit-specific toxin effects, also taking into account that these toxins bind at subunit interfaces.

In order to fully address this revision point, the following details were provided upon our request.

i) the comments made by reviewer #2 (i.e the request to test the toxins on recombinantly expressed nAChRs to support claims on subunit specificity)

ii) A more general request to revise the language to account for the fact that the toxins bind to subunit interfaces. In other words: we ask the authors to be more explicit on whether they conclude that the toxins bind to the interface of two identical subunits

(homomeric α5-α5 / α6-α6 / α7-α7 interfaces) or requires only a single principal (+) (or complementary (-)) interface of either of the toxin-binding subunits (α5, α6 or α7).

Please feel free to reach out if anything else requires clarification.

From our mass spectrometry data we cannot conclude whether a homomeric or a heteromeric receptor consisting of Dα5, Dα6, Dα7 binds to α-Btx. However, as we note in the manuscript, since we recover α-Btx bound receptors in each of the mutant pull-downs it is unlikely that a single receptor subtype is responsible. We include new Western blot data with tagged Dα6 that indicate at least one binding receptor contains a Dα6 dimer (Figure 3figure supplement 1C, 1D). These dimers are lost by adding dithiothreitol after performing pull-downs with α-Btx affinity beads (Figure 3C). We also note that Lansdell and colleagues show in an S2 expression system that a homomeric Dα6/5HT3A chimeric complex is able to bind on the cell surface to α-Btx, this aligns with our finding (Lansdell et al., 2004). Testing recombinant nAChRs in Xenopus laevis for α-Btx goes far beyond our in vivo analysis, which we note is well supported by cited data from heterologous or cell line systems. We emphasize that both our genetic and biochemical data are consistent with our claims and support the view that the Dα5, Dα6 and Dα7 subunits are involved in binding.

2) Additional negative stain EM analysis to substantiate claims with adequate statistics, or adjust analysis and claims to reflect limitations.

We performed additional negative stain EM analysis and provide our reproduced TEM micrographs (Figure3-Source Data1). Further images were taken from purified nAChRs fractions. Membrane pellets were solubilized with SMA and these fractions were used to enrich for nAChRs with pull-downs using α-Btx affinity beads. TEM micrographs of negative stain grids were taken from three independent biological replicates using w1118, thus total we provide 28 images (15 micrographs from purified nAChRs with α-Btx affinity resin and 13 micrographs from total SMALP fractions before enriching nAChRs with α-Btx, Figure3 Source Data1). Counting individual particles or structures in the micrographs is not possible due to the complexity and heterogeneity within the samples.

From the newly acquired images it has become clearer that the nAChRs seem to bind to the TEM film grids in a preferred orientation providing mostly top/bottom views. This makes it much harder to visualize side views of the nAChR receptors as they are rarely positioned on the grids in that orientation. The heterogeneity within the samples and the preferred orientation of the nAChRs on the TEM grids does not allow to record a sufficient number of TEM micrographs in a certain orientation to undertake 2D class averaging (Gallagher et al., 2019).

3) Data on alpha5/7 localisation or a clear narrative and justification of why only alpha6 was subjected to localisation studies.

We have provided more detail on why Dα6 was used in the localization studies. In brief, there are several studies reporting Gal4 driven Dα6 localization and RNA localization data, giving us information to validate our novel endogenous gene tagging approach. We emphasize that this was a pilot study to test the feasibility of the N-terminal tagging approach and in support of this provide a supplementary figure with preliminary data tagging three other subunits.

In addition, Dα6 is a well characterized target of the Spinosad insecticide (Zimmer et al., 2016; Martelli et al., 2022), which meant our tagged line was likely to have broader relevance.

4) Language should be carefully revised based on the comments below to make it accessible to a broader audience. Additionally, the manuscript needs to better reflect prior work, provide greater clarity on the final genotypes of the knockout strains and provide additional context and motivation for the proteomic/glycosylation work.

We have edited all the comments below as best as we can to improve understanding. In addition, we have incorporated earlier findings and increased citations to prior work to better contextualise our findings. We provided more clarity on the phenotype of our newly constructed knockout strains, improved figures, and added a detailed motivation for our proteomic/glycosylation work.

A straightforward way to address these requests would be inclusion of additional results. Further details of these concerns are in the reviews, along with suggestions that we hope you will find helpful.Reviewer #1 (Recommendations for the authors):Further comments and suggestions to the authors:1) One is left to wonder why the authors didn't run the solubilized sample on an SDS page gel and subject it to Western blotting with nAChR specific antibodies to show how α-Btx affinity resin helped enrich the sample with nAChR. This would have been more quantitative than their TEM analysis.

We addressed this request by performing an immunoblotting experiment with an mVenus tagged Dα6 nAChRs subunit (Figure 3 C, Figure 3—figure supplement 1B).

2) The authors should provide info on the amount of membrane (g) used in their SMA solubilization described in the methods section.

We have clarified this and provide an additional table of four biological replicates (Table 9).

3) The authors should substantiate their TEM analysis and claims with statistics by collecting more than a single TEM micrograph and used TEM software to calculate 2D class averages and compare those to see if there are "side" and top views of nAChR. The authors should also provide details on how many images their analysis is based upon in the methods section.

We performed additional negative stain electron microscopy (EM) analysis of three biological replicates. Requested additional EM experiments were performed with SMALP samples which were enriched with α-Btx affinity beads. Moreover, additional micrographs from SMALP samples without enriching them for nAChR were formed as requested. By repeating our analysis, we found increased numbers of ring-like structures in samples enriched for nAChRs that were not evident in unenriched preparations. Close inspection of the micrographs revealed no other obvious recurring structures.

It is possible be that nAChRs preferentially bind to TEM film grids in a certain orientation during negative staining, an issue known with other receptors and proteins. One reason for this could be the upper surface of the carbon grid, which allows the particles to only turn in a certain direction (Gallagher et al., 2019).

We describe in the methods section how many micrographs we used in the study.

4) Figure 3C which validates coupling of α-Btx to the resin rightfully belongs in a lab notebook, not as a figure in the main text.

We removed and replaced Figure 3C with Western blotting data.

5) The authors should include a description of how to calculate the hydrophobicity score.

We included a detailed description how we calculated the hydrophobicity score:

“The hydrophobicity score was calculated by the sum of the hydrophobic or hydrophilic properties of amino acids divided by the length of identified proteins.”

6) Line 308. As a reader, one wonders why there is no evidence of peptides mapping to TMH domains. The authors should explain.

We included a reason in the main text:

“However, we found no evidence of peptides mapping to the TMH domains, which was not unexpected since peptides from these regions contain hydrophobic amino acids that are known to be difficult to detect by mass spectrometry (Carroll et al., 2007).”

7) Lines 308-310. The authors should expand their description of the mapping of the identified peptides. It is not clear why the identified peptides are outside of the α-Btx binding sites with the information that is currently given.

We included reasons:

“This may indicate that α-Btx binding sites are not structurally conserved or that binding of α-Btx alters local proteolytic susceptibility and as a consequence peptides identifiable by mass spectrometry are not generated.”

8) A Figure like the one in Supplement 3B should be found in the main text figures and not in the supplement. However, the figure should be better annotated such that the reader can easily understand whether the α-Btx is docked into the model or part of it.

We included supplement 3B into the main text of figure 5. Superimposed nAChR structures of *Homo sapiens* and *Torpedo californica* α-subunits are now clearly annotated within figure 5C.

Further annotations are now included in figure 5D showing positions of identified peptides of *D. melanogaster* ligand-binding domain in homology regions. Structure of α-Btx which docks into the model is now well-annotated within figure 5D.

9) Figure 6B. It is not clear which flowthrough that is meant. From α-Btx resin or the HILIC resin.

We changed the Figure 6B and Figure 6B legend and added that peptides were measured in the flow through of the HILIC resin.

10) Information on how the -SMALP sample was prepared is missing from the methods. Was it simply prepared without SMA in the buffer or was something different done?

We have added the following sentence to the material and methods section which describes that samples were prepared also without SMA in the buffer:

“As a comparison for the proteomic approach membrane pellets were solubilized in 5 %

SMALP solution without adding the copolymer into the buffer.”

Reviewer #2 (Recommendations for the authors):This study was carefully and comprehensively conducted to demonstrate that the two peptide toxins differentially target insect nAChRs by in vivo and biochemical experiments. It is, however, noteworthy that the in vivo data may be the results of compensation by other nAChR subunits. In the paper the ventral nerve cords of adult flies were used to purify the proteins but when other preparations or parts are used, different results may be obtained. The pull-down assay identified the main targets, but in the case of α Btx, alpha3 was also identified but in much lower frequency, which should also be noted in the text as a fact. As regards localisation of the subunits, it is plausible to investigate the localisation of α5 and α7 subunits to examine if these subunits are colocalized with α6 to form heteromers as the toxin binding sites.

We note in the manuscript that Lansdell and colleagues found after co-expressing a Dα3 subunit together with a non-α subunit such as Dβ2, or mammalian β2, γor δ, α-Btx binding was detectable in heterologous S2 expression systems (Lansdell et al., 2012). After our enrichment of nAChRs to study glycosylated peptides of receptor subunits we identified Dα3 and Dβ3. These two subunits were only identified after deglycosylation of enriched peptides with Endo H but not with a PNGase F treatment suggesting that these subunits are modified and are difficult in their identification.

Additional Western blotting investigations were performed and suggest that two Dα6 subunits interact with α-Btx (Figure 3—figure supplement 1 C, 1D). This may indicate that two Dα6 subunits are next to each other in a nAChR complex. From our mass spectrometry data we cannot rule out if Dα5, Dα6 and Dα7 each form a homomeric α5-α5 / α6-α6 / α7α7 interface that binds to α-Btx or if there are also other heteromeric possibilities feasible.

Given that none of the mutants are completely resistant to α-Btx and that pull downs in each of the mutants still have α-Btx binding, it suggests to us that α-Btx binding receptors can exist in a number of conformations.

Reviewer #3 (Recommendations for the authors):The use of the term deletion to describe the strains is not quite accurate. The genes are clearly disrupted by the constructs and so a more appropriate term along these lines could be considered.

We have replaced the word deletion to mutation or to null.

For the injection assay results, presentation of the individual replicate data points on the figure 2 would be a clearer way to provide insights to the reader regarding variability rather than relying on the table. I also have a question about the description of the figure (line 190) of there being 10 larvae per replicate, however there are a number of data points where 25% is reported in Appendix Table 2. How was this calculated and would raw data be more suitable to report in the table given the number of larvae tested was 10 for each replicate?

As requested we have included individual replicate data points in figure 2. Moreover, we address the following requests by testing three independent replicates with a total of 10 larvae per genotype and corrected the legend of Figure 2:

(A) Bar graph of the survival rate, measured as the percentage of pupae formed, following larval 189 injection of 2.5 nmol/g Hv1a in the indicated homozygous lines. **P=0.0035 (oneway ANOVA (F(11,24)=4.99, P=0.0005 with Bonferroni’s multiple comparisons test). Mean ± SD of 3 independent replicates in each group (10 injected larvae in total). Individual replicate data points are shown (B) Survival rate following larval injection of 1.25 nmol/g αBtx. **P<0.001, ***P=0.0001 (one-way ANOVA (F(11,24)= 7.921, P<0.0001, followed by Bonferroni’s multiple comparisons test). Mean ± SD of 3 independent replicates in each group (10 injected larvae in total). w1118 is the wild-type base stock, THattp40 and THattP2 are the Cas9 lines used to establish the mutants, w1118 + PBS represents the injection control.

Related to the point in the public review, if the Dbeta1 strain in this study is able to survive until pupation, would it be possible to conduct the injection experiments on these to assay for the involvement of Dbeta1, particularly in the action of Hvl1a? This would enable results from the full set of genes to be reported, strengthening the findings.

As we indicated above, we were unable to recover any homozygous larvae.